# Platelets regulate neural and oligodendroglial progenitors when infiltrating the brain parenchyma
Christina Dimitriou [1,10,11,13], Maria Anesti[1,12,13], Stefanos Kaplanis[2], Theodora Mourtzi[1], Aggeliki Dimopoulou[1], Dimitrios Dimitrakopoulos[1], Filippos Katsaitis [1], Maria Nousia[1], Konstantina Mastori[1], Kyriaki Karassavidou[1], Efthimia Aleiferi[1], Dimitrios Lagogiannis[1], Esra Tahir[3], Amber R. Philp[4], Adamantia Kouvela [5], Constantinos Stathopoulos[5], Domna Karagogeos [2], Robin J. M. Franklin[6,7], Cedric Ghevaert[8], Francisco J. Rivera [4,9] & Ilias Kazanis [1,3,6] ✉

The mammalian, including the human, postnatal brain harbors distinct neurogenic and oligodendrogenic progenitor populations, whose activity during homeostasis and degeneration is closely related to blood vessels. Here, we investigated the regulatory role of platelets, that are easily and unlimitedly accessible and can modulate tissue regeneration. In co-cultures, platelets created a pro-stemness environment for neural stem/progenitor cells (NSPCs) and enhanced the differentiation of oligodendrocyte progenitor cells. The effects were dependent on platelet numbers and the culture microenvironment, while the molecular signature of the NSPC response confirmed the role of Platelet-Derived Growth Factor and the upregulation of stemness-promoting genes. The same effects were exerted in vivo, but only when platelets were injected directly into the striatal parenchyma or were extravasated, in severely thrombocytopenic mice. In contrast, thrombocytopenia without platelet tissue-infiltration, in Nbeal2 knockout mice, did not affect the brain's progenitors. Our results, demonstrate the potential value of platelets in neuro-regenerative strategies.

In the mammalian postnatal brain, different populations of stem and progenitor cells are found in different locations, serving distinct functions[1]. Multipotent neural stem cells cluster within specialized microenvironments, called stem cell niches, where they infrequently exit quiescence to generate neurons and glia via intermediate, lineage committed, progenitor cells[2–4]. There are two well-described niches in the rodent and the human brain: the subgranular zone of the dentate gyrus of the hippocampal formation and the subependymal zone (SEZ, also known as ventricular-subventricular zone) of the lateral walls of the lateral ventricles[5]. Even though the functional outcome of these two niches is different, with progeny of neuronal/astroglial lineage in the hippocampal niche contributing to memory and learning[6–8], and progeny of neuronal/astroglial/oligodendroglial lineage in the SEZ contributing to olfaction[9], tissue regeneration[10] and myelination[11], one common feature of their structure is the proximity of Neural Stem and Progenitor Cells (NSPCs) to blood vessels. The vasculature of the SEZ has a specialized architecture, with a blood vessel network that is denser and reaches the ventricular wall at shorter distances when compared to other periventricular areas[12,13], with low blood-flow rate[12,14], increased levels of leakiness[15], and with endothelial cells that play a direct regulatory role on

NSPCs[16]. Another pool of brain progenitors consists of Oligodendrocyte Progenitor Cells (OPCs), that are scattered throughout the parenchyma, retaining a life-long mitotic activity and contributing to the generation of new oligodendrocytes, capable of myelinating and remyelinating axons, both in the grey and the white matter[17]. OPCs dynamically maintain their numbers and distribution[18], with their microenvironment and the vasculature playing a significant role[19–22]. Latent neural progenitors have also been identified away from the neurogenic zones, for example, in the cortex and the striatum, in the rodent and the human brain[23–25], but the microenvironment and systemic factors that contribute to their regulation have not been fully elucidated.

Platelets (or thrombocytes) circulate in high numbers within blood vessels and have a fast rate of turnover[26]. Although they are anucleate cells, they are loaded with multiple signaling molecules, such as proteins and micro-RNAs, mostly packed within granules[27–29]. They are rapidly recruited to areas of vascular injury, where they interact with endothelial cells, with each other, and with cells of the immune system to contribute to blood clotting and vessel healing[30,31]. Even though platelets are not normally present within the tissue parenchyma, their cargo is consistent with an

---

ability to act upon a range of cell types[32,33], including those of the nervous system[34–38], and they can infiltrate the parenchyma in cases of hemorrhages.

We have previously shown that platelets accumulate in the SEZ vasculature in response to a focal injury at the proximal corpus callosum (CC)[39] and that circulating platelets act as regulators of OPCs in areas of demyelination[40]. Additional evidence suggests that circulating platelets can affect NSPCs of the hippocampal niche[36,38], and that pro-neurogenic effects can be achieved after their intracerebroventricular administration[41]. Here, using a range of approaches, we identify the key effects, and the underlying molecular signature, of platelets on NSPCs inside and outside the SEZ niche as well as on OPCs, and we demonstrate that these effects are exerted only by extravasated platelets. This work advances the potential to use this naturally unlimited, and suitable for autologous use, source of cells in regenerative biomedicine.

## Results

### Platelets create a pro-stemness microenvironment for NSPCs and a pro-differentiation microenvironment for parenchymal OPCs, in co-cultures

We have previously shown that platelets accumulate in the vasculature of the mouse SEZ neural stem cell niche after an experimental focal demyelination at the proximal CC[39]; an injury leading to recruitment and migration of SEZ cells at the area of lesion[42]. This selective presence of platelets in the niche prompted us to investigate their possible effects on NSPCs and we explored this, firstly in vitro, by co-culturing washed, non-activated, mouse platelets, isolated from the *vena cava inferna*, with mouse NSPCs isolated from the SEZ (Fig. 1A–E). We confirmed that platelets remained inactivated during isolation, using FACS or immunocytostainings on coverslips, for CD41, a universal platelet marker, and CD62P (P-selectin), a marker of activated platelets (Supplementary Fig. 1A–F). Each biological NSPC sample was split into sub-cultures, and different numbers of platelets (collected by the same blood samples), at a ratio of "NSPC:platelets" that ranged between 1:40 and 1:1200, were seeded directly on NSPCs (which are characterized by the expression of NESTIN and of SOX2) (Fig. 1B–E, H–J and Supplementary Figs. 1G and 2C). Furthermore, a "no platelets" culture serving as the internal control within each experiment. Co-cultures were maintained for 3 days, without any media changes, in NSPC proliferation or differentiation medium (i.e., in the presence or absence of FGF2 and EGF, respectively). By assessing the percentage of CD41+ platelets co-expressing CD62P, it was revealed that platelets became activated in vitro and that in the presence of NSPCs, their activation was significantly lower than when cultured on their own (Supplementary Fig. 1H), indicating that these cell groups were interacting with each other. In a separate approach, measuring mitochondrial activity with the MTT assay[43], we were able to confirm that platelets became activated when kept in NSPC media, with their activation being very fast in low densities (relevant to NSPC:platelets ratios up to 1:40) and with platelets becoming fully exhausted after 5 days in culture (Supplementary Fig. 1I).

We analyzed the immunoreactivity for the key NSPC transcriptional factor SOX2, for Ki67 or PCNA to detect proliferating cells, as well as for DCX and OLIG2 to identify cells of neuronal and oligodendroglial identity, respectively, in NSPC differentiation conditions. The presence of platelets was not toxic, as numbers of surviving cells remained at control levels (Fig. 1F). However, the percentage of SOX2 immunopositive cells was significantly increased across all ratios (Fig. 1G). Levels of proliferation, that are markedly low in differentiation conditions, showed a dichotomous effect: the 1:40 NSPCs:platelets ratio resulted in significantly reduced levels of Ki67, while the 1:1200 ratio led to a 3.5 times, significant, increase (Fig. 1K), with approximately 5 to 10% of cells on day 3 co-expressing Ki67 (Fig. 1K, L dark blue bar). Notably, in the highest tested presence of platelets, levels of proliferation became non-statistically different from the levels of proliferation we observed in pure NSPC cultures with cells grown in proliferation conditions for 1 or 3 days (Fig. 1L). The positive effect of high numbers of platelets to NSPC proliferation, in conditions normally restricting mitosis, was also observed by PCNA immunocytochemistry,

with the percentage of PCNA+ cells showing a strong but not statistically significant trend of increase, up to 4.2 times (Supplementary Fig. 2A). Cell fate commitment towards the neuronal and the oligodendroglial lineage was not affected (Supplementary Fig. 2B–G). Interestingly, the co-culture of NSPCs and platelets in proliferation conditions had no effects on NSPCs, apart from a notable, but not statistically significant, increase in the percentage of Olig2+ cells at the 1:1200 ratio (Supplementary Fig. 2E–G).

To assess the possible effects of platelets on the other major population of brain progenitors, those of the oligodendroglial lineage, OPCs were isolated from the brains of newborn mice and were co-cultured with platelets for 4 days in conditions that support the differentiation of OPCs to oligodendrocytes, a process monitored by the expression of the oligodendrocyte markers CC1 and MBP (Fig. 1M–Q). Because OPCs can be exposed to high numbers of platelets in areas of acute demyelination, in patients with multiple sclerosis[44,45] and in relevant animal models[40,46], we assessed OPC:platelet ratios at the higher end of those assessed for NSPCs (1:1000 and 1:3000). MBP immunoreactivity was significantly increased (2.4 times) only in the highest presence of platelets (Fig. 1P). The numbers of both CC1+ and MBP+ cells were also found to be significantly increased in the presence of 3000 platelets per OPC (Fig. 1Q). It is worth noting that the percentage of the more mature oligodendrocytes (those co-expressing MBP and CC1), were similar in both conditions (at approximately 60% of CC1+ cell), indicating that platelets affected the initial stages of differentiation and that the rate of maturation remained similar (see dotted arrows in Fig. 1Q).

Together, our in vitro data revealed that platelets are able to create a pro-stemness microenvironment for NSPCs, enhancing (by maintaining or re-inducing) the expression of SOX2 and supporting mitosis, and to promote differentiation of parenchymal OPCs.

### Extravasated platelets act on NSPCs and induce Sox2 expression

To complement our previous observation of the accumulation of platelets in the SEZ vasculature, in response to demyelination in the proximal CC[39], we analyzed, by means of immunofluorescence for laminin (to mark blood vessels) and CD41 (to mark platelets), other models in which NSPCs have been experimentally shown to be affected, via traumatic or pharmacological stimuli. In a rat model of cerebral stroke, analyzed 4 weeks after an 1 h, unilateral, middle cerebral artery occlusion, the area of lesion was identified in the affected hemisphere via the increased deposition of laminin on blood vessels and in the parenchyma (the tissue remained unaffected in the contralateral hemisphere, Fig. 2A, B)[47]. The lesion included the striatum of the occluded hemisphere (Fig. 2B1), but also parts of the SEZ (Fig. 2B2). We have previously reported that NSPCs are activated only within these niche domains[47] and platelets were found to accumulate selectively inside and outside the blood vessels of these areas (Fig. 2B). On the other hand, one week after the bilateral intracerebroventricular injection of neuraminidase, a toxin that leads to structural damage of the niche due to the denudation of the ependymal cell layer, but we have previously reported not to functionally affect NSPCs[48], platelets were found to accumulate only within the blood vessels of both hemispheres' SEZs (Fig. 2C, D). Finally, we looked in the Substantia Nigra (SN) of mice that had received, intraperitoneally, during postnatal days 14 to 60, the micro-neurotrophin BNN-20 (or saline) (Fig. 2E, F), which activates neurogenesis in the SN[49]. The administration of BNN-20, and not of saline, in the absence of any traumatic or other inflammatory stimuli, led to the emergence of platelets anchored within the vasculature and infiltrating the SN parenchyma (Fig. 2F1–F10). Together, these data revealed that platelets are found outside the brain's vasculature in experimental conditions of recruitment of NSPCs (MCAO, neurotrophin) and not when NSPCs are not activated by local injury (neuraminidase).

Our histological and in vitro analyses strongly indicated that platelets could exert an effect on NSPCs when located outside the blood vessels. To consolidate this assertion, we injected washed platelets that were labeled with the lipophilic dye PKH26, to enable their visualization, directly into the brain parenchyma. We first performed co-cultures of PKH26-labelled platelets and NSPCs (Fig. 3A–C), and we found no indication of altered

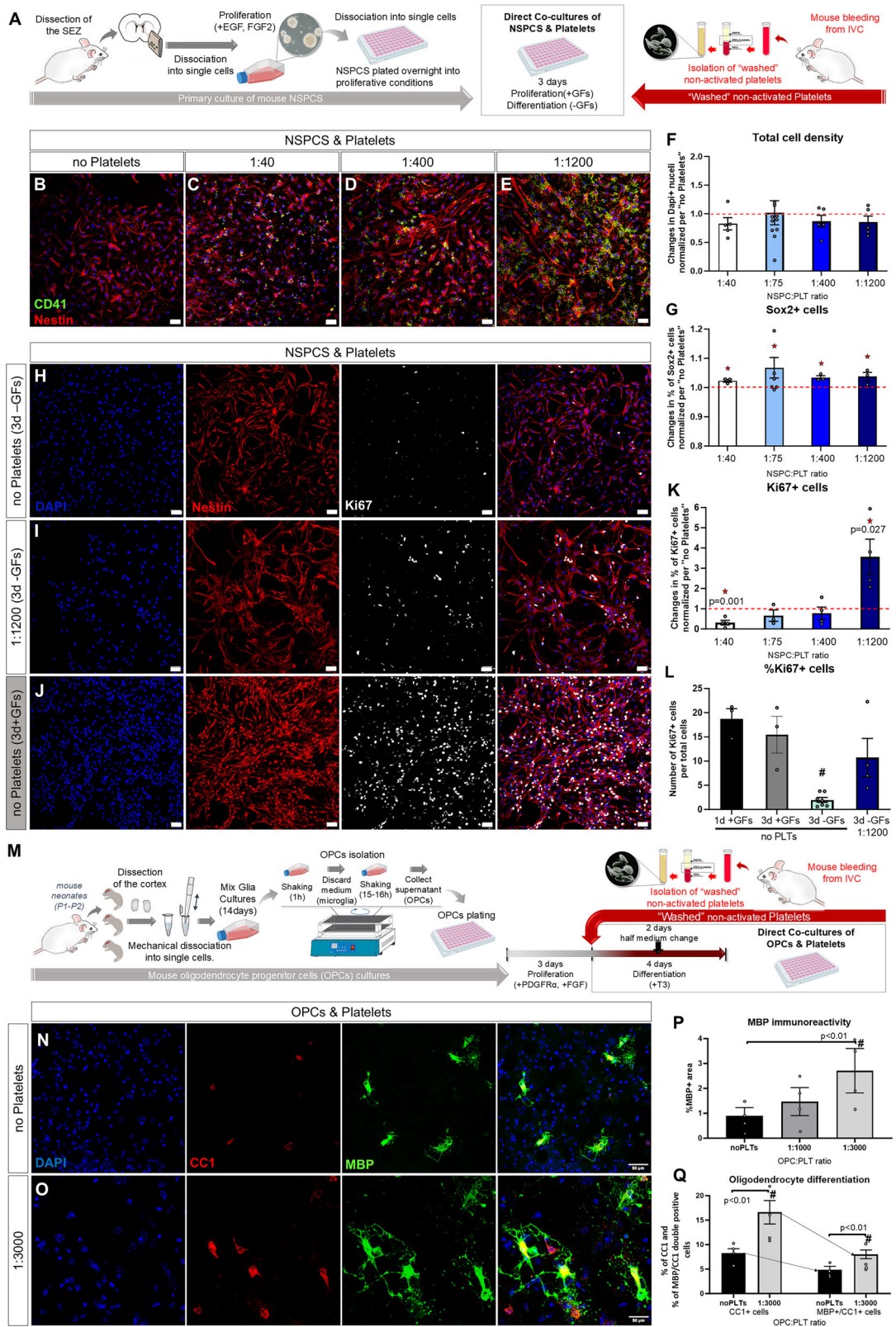

biological function caused by the dye. PKH26-labelled platelets showed no toxicity and led to a significant increase in the percentage of SOX2+ cells (Fig. 3D, E). Subsequently, approximately 30000 platelets were unilaterally injected into the striatum, a structure harboring latent neural progenitors in the mouse brain[25] (Supplementary Fig. 3A, B) or at the SEZ dorsal horn (Supplementary Fig. 3L, M), with DMEM injected into the same areas of the contralateral hemisphere. The injection dose was selected as it constituted the maximum number of platelets that could be concentrated within 1 µl of medium without visual signs of activation (e.g., clotting). Based on tissue cell-densities calculated in previous experimental work[42] and on further pilot calculations, we estimated an average cell density of $6 \times 10^6$ cells/mm$^3$ (total cells, including neurons, glia, and endothelial cells). Considering that

**Fig. 1 | Co-culture of platelets with NSPCs and OPCs. A**, **M** Schematic illustrations of the experimental plans. **B–E**, **H–J** Confocal microscopy microphotographs of NSPC cultures without platelets (PLTs, as in **B**, **H** and in **J**) and of NSPCs co-cultured with PLTs at different ratios, after immunostaining for KI67 (to mark proliferating cells, in white), CD41 (to mark PLTs, in green), and NESTIN (to mark NSPCs, in red). Note that the culture shown in (**J**) is in proliferation conditions. **F**, **G**, **K** Graphs showing the changes in the total number of cells (in **F**), or in the percentage of Sox2+ (in **G**), and Ki67+ (in **K**) cells, after co-culturing NSPC and PLTs at different ratios in differentiation conditions. The scatter plot bars show mean values and error bars the SEMs. The value of each NSPC biological sample (formed by 1 to 3 technical replicates) is depicted with a circle and is normalized per the "no-platelets" culture of the same sample, which is at "1.0" and is indicated by the red, dotted line. **L** Graph showing the percentage of Ki67+ cells after immunostaining of pure NSPC cultures in the presence or absence of growth factors (GFs), for 1 or 3 days and in NSPC/PLT

co-cultures at a 1:1200 ratio for 3 days without GFs. **N**, **O** Microphotographs of OPC cultures (in **N**) and of OPC/PLT co-cultures (in **O**), after immunostaining for CC1 (in red) and MBP (in green). **P**, **Q** Graphs showing the percentage of MBP+ immunoreactivity (in **P**) and the percentage of CC1+ and MBP+ cells, per optical field, in pure OPC (no PLTs) cultures and in OPC/PLT co-cultures at different ratios. Note that all MBP+ cells co-express CC1 and that the fraction of CC1+ cells that have matured to also express MBP (indicated by the arrows) is similar (at approximately 60%) irrespectively of the presence of platelets. [Statistical analyses were performed with 1-way ANOVA, followed by the Tukey post-hoc analysis. Solid, red, stars indicate a significant ($p < 0.05$) change compared to "no PLTs". The hash mark in (**L**) indicates significant difference compared to all the other conditions and in (**P**, **Q**) highlights the statistical differences indicated by the horizontal brackets. Scale bars: 50 μm] [Numeric data available in Supplementary Data 1].

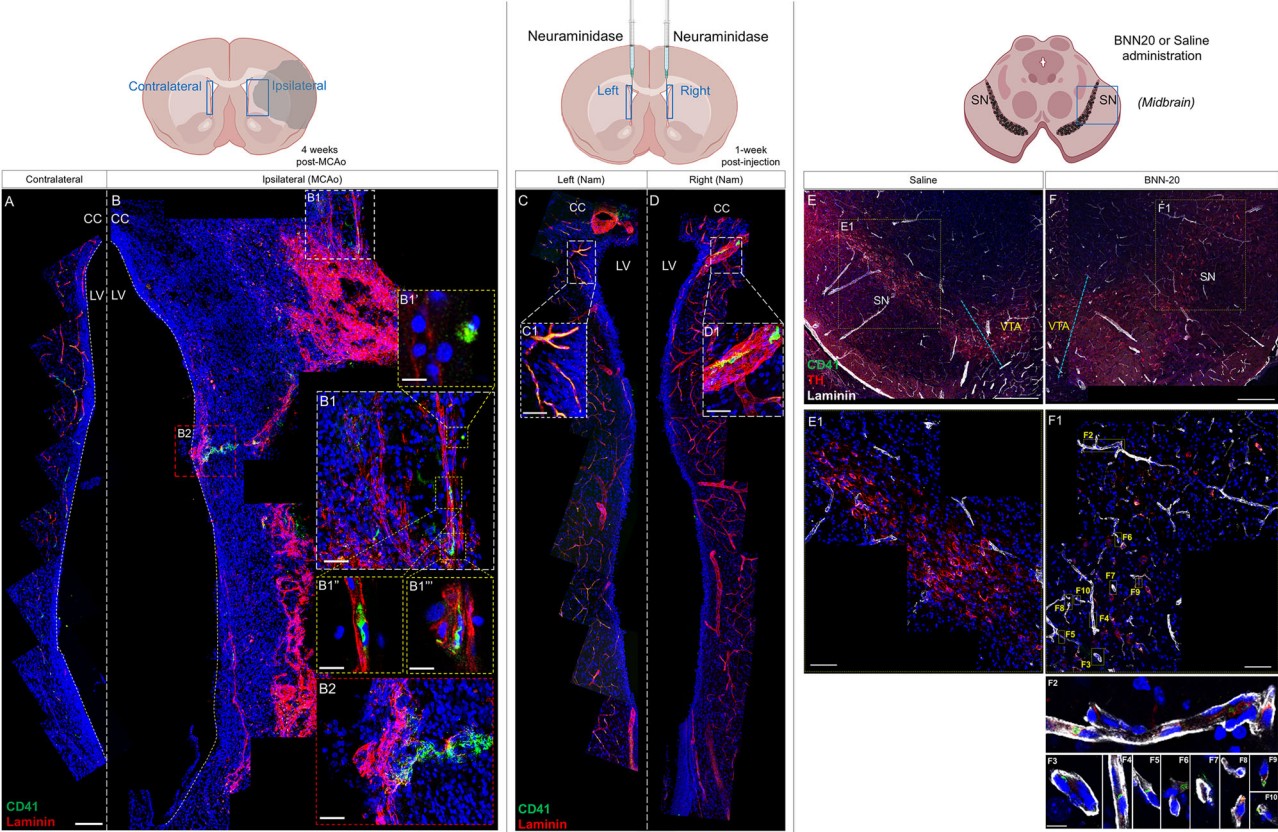

**Fig. 2 | Presence of platelets in brain areas of NSPC activation.**
**A**, **B** Microphotographs of the area next to the lateral ventricle (LV, the walls of which are indicated by white, dotted, lines) in rat brain sections, 4 weeks following one hour of a unilateral, Middle Cerebral Artery Occlusion, after immunostaining for laminin (in red) and CD41 (in green). The area affected by stroke is shaded in the illustration located above the photographs and the areas shown are outlined by blue rectangles. Note the absence of CD41+ platelets and the limited laminin immunoreactivity at the LV wall in the contralateral hemisphere, in contrast to the high expression of laminin in the striatum (for example, in the area outlined in **B1**) and in parts of the SEZ, positioned at the lateral wall of the LV (for example, in the area outlined in **B2**). Platelets are found outside (**B1'** and **B1'''**) and inside (**B1''**) of

laminin+ blood vessels in the striatum, and outside blood vessels, in the parenchyma, in the SEZ. **C**, **D** Microphotographs of the area next to the LV in rat brain sections, one week after the bilateral intracerebroventricular injection of neuraminidase (nam). Note the presence of CD41+ platelets aggregated within laminin + blood vessels (shown in magnification in **C1**, **D1**). **E**, **F** Microphotographs of the Substantia Nigra (SN) in mouse brain sections following the intraperitoneal administration of saline, or of the microneurotrophin BNN-20, after immunostaining for dopaminergic neurons (TH+ in red), laminin (in white), and CD41 (in green). Note the presence of CD41+ platelets within (**F2–F5**) and outside (**F6–F10**) laminin+ blood vessels after the administration of BNN-20. [scale bars: 100 μm in low magnification images; 20 μm in high magnification insets].

the injection of 1 μl of platelets/DMEM could directly affect a maximum volume of 0.5 mm³, the injection of 30000 platelets would result in a cell:platelet ratio of 100:1, although this ratio would be closer to the ratios tested in vitro (such as 1:100), more proximal to the site of injection. Mice were killed 4 days after the injections and tissue was immunostained for SOX2 and PCNA (Fig. 3F–I). PKH26-labelled platelets were visible only at the site of injection, with many of them located at the surface of blood vessels and, in

few cases, having entered within the local vasculature (Supplementary Fig. 3A–I). In the SEZ area, the presence of SOX2+ and SOX2+/PCNA+ cells was not significantly affected by the injection of platelets (Fig. 3J). In the striatum, we also found similar numbers of PCNA+ cells around the site of injection of DMEM or platelets. Notably, though, at the area of platelet injection, significantly more SOX2+ and SOX2/PCNA double-positive cells were detected (Fig. 3M). To investigate more the identity of proliferating

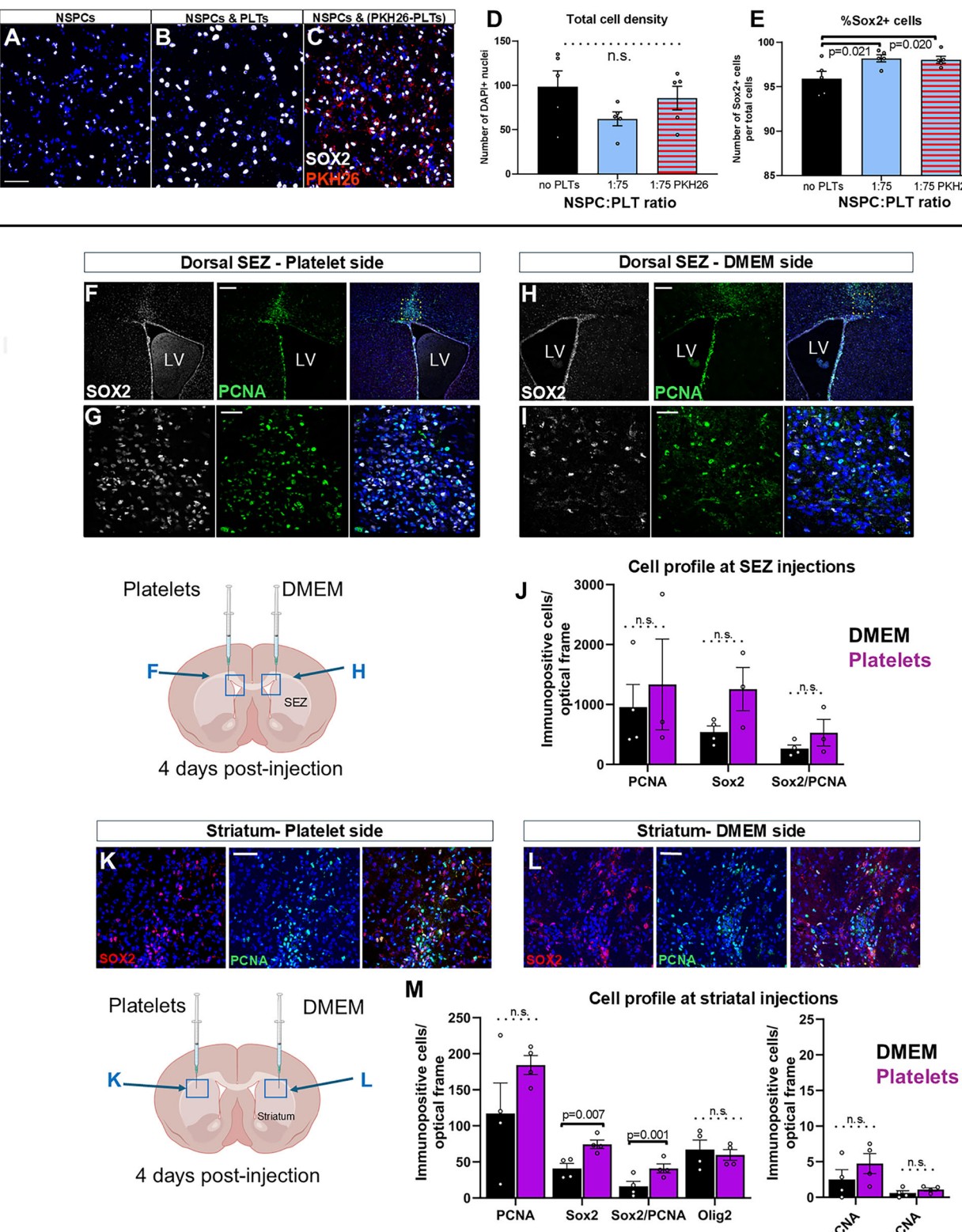

cells at the sites of injection in the striatum, we performed additional immunostainings for Olig2, GFAP (to mark astrocytes), Iba1 (to marks microglial cells), and PCNA or phosphorylated-Histone 3 (PH3) (Fig. 3M and Supplementary Fig. 3J, K). We found only small numbers of mitotic Olig2+ and GFAP+ cells and no double positive Iba1/PH3 cells (Fig. 3M and Supplementary Fig. 3J, K). These results suggest that the majority of

PCNA+ cells in both DMEM- or platelet-injected striata were of peripheral origin (e.g., macrophages), a finding consistent with the presence of a compromised vascular integrity, as judged by the presence of platelets having entered local blood vessels (Supplementary Fig. 3C–I). The observation that the direct injection of platelets in the striatum results in the emergence of SOX2+ cells, a fraction of which are mitotic and are not

**Fig. 3 | Intracerebral injections of platelets. A–C** Microphotographs of a pure NSPC culture (in **A**) and of NSPC/platelets (PLTs) co-cultures (in **B, C**), after immunostaining for Sox2 (in white). PKH26-labelled platelets are visible in red, in (**C**). **D, E** Graphs showing the total cell numbers and the percentage of Sox2+ cells in pure NSPC cultures (no PLTs) and in co-cultures of NSPC and PLTs, at a 1:75 ratio, to compare the effects of washed PLTs with or without labelling with PKH-26. **F–I** Microphotographs of mouse brain sections, immunostained for PCNA (in green) and Sox2 (in white), 4 days after the intracerebral injection of PLTs, or DMEM, in the dorsal horn of the SEZ, as depicted in the adjacent schematic illustration. The areas outlined in (**F**) and (**H**) are shown in higher magnification in (**G**) and (**I**), respectively. **J** Graph showing the numbers of PCNA+, Sox2+, and double + PCNA/Sox2 cells at the area of injection of DMEM or platelets, in the dorsal SEZ.

**K, L** Microphotographs of mouse brain sections, immunostained for PCNA (in green) and Sox2 (in red), 4 days after the intracerebral injection of PLTs, or DMEM, in the striatum, as depicted in the adjacent schematic illustration. **M** Graphs showing the numbers of PCNA+, Sox2+, double+ Sox2/PCNA, Olig2+, double+ Olig2/PCNA, and double+ GFAP/PCNA cells at the area of injection of DMEM or platelets, in the striatum. [scale bars: 30 μm in (**A–C, G, I, J**) and 100 μm in (**F, H**). Bars show mean values and error bars the SEMs. 1-way ANOVAs, followed by Tukey post-hoc analyses were performed in (**D, E**). Paired student *t*-tests, for each marker set, were performed in (**J, M**). n.s. not significant and statistical differences are depicted by horizontal brackets and the respective *p*-values. LV Lateral Ventricles] [Numeric data available in Supplementary Data 1 and 2].

co-expressing Olig2 or GFAP, suggests that platelets can induce the recruitment of local, parenchyma neural progenitors.

## Tissue-infiltrating platelets positively affect oligodendroglial lineage cells

To assess further the role of platelets as in vivo systemic regulators of the brain's NSPCs, we induced a unilateral, focal, demyelinating lesion in the CC and analysed both the SEZ and the CC in normal and thrombocytopenic mice (Fig. 4A). Long-term, moderate, thrombocytopenia was induced by knocking out the expression of the Nbeal2 gene (Nbeal2-KO), which leads to a 50% reduction in the number of circulating platelets that also lack α-granules[50]. Short-term, severe (>90% depletion), thrombocytopenia was induced via the depletion of circulating platelets for 4 days, by two intra-peritoneal injections of the a-CD42b antibody (Fig. 4B)[40]. Because in Nbeal2-KO mice, thrombocytopenia is not full, we assessed the presence of platelets within the SEZ vasculature ipsilateral and contralateral to the lesion. In the non-injured hemisphere, we detected a fraction of blood vessels with adherent CD41+ platelets (Supplementary Video 1), a phenomenon possibly facilitated by the low rate of blood flow[14] observed in the SEZ, that was not different between the genotypes. However, in response to injury, the fraction of blood vessel fragments with adherent platelets was significantly increased in wild-type (WT) mice, a phenomenon that was absent from Nbeal2-KO mice (Fig. 4C, D).

Lysolecithin, a demyelinating agent, was injected into the CC near to the SEZ. Tissue was analysed at 5-, 7-, and 14-days post-lesion (dpl) in WT and in Nbeal2-KO mice and at 7 dpl after chemically-induced thrombocytopenia (Figs. 4A and 5A). The effects of lysolecithin were initially assessed by proteolipid protein (PLP), a key myelin component, immunoreactivity (Fig. 4E–L), which was found to be reduced to 60% and 40% of its normal levels (the respective CC area in the contralateral, non-lesioned hemisphere served as internal control) at 7 dpl and 14 dpl, respectively, irrespective of experimental group (Fig. 4M). New myelin starts to be generated by OPC-derived oligodendrocytes after 14 dpl and the area is fully remyelinated after 30 days[42].

We quantified the cell density and the presence of PCNA, DCX, and OLIG2 immunopositive cells in the SEZ (Supplementary Fig. 4A, B and Fig. 5B–E). All PCNA+ cells within the niche co-express the transcription factor SOX2[13] (Supplementary Fig. 4C); thus, they were considered to be NSPCs. Focal demyelination in the CC did not affect the total cell density in the WT SEZ (Fig. 5B, black bars), although it led to a transient, significant, increase in the percentage of PCNA+ cells at 5 dpl (Fig. 5C); thus, indicating that the injury in the CC affected the SEZ niche. The pools of OLIG2+ and DCX+ progenitors were not affected (Fig. 5D, E, black bars). Long-term, moderate, thrombocytopenia had no effects in the SEZ (Fig. 5B–E, cyan bars). However, short-term, severe, thrombocytopenia led to a significant increase in the percentage of OLIG2+ cells, both in the control SEZ and at 7 dpl, compared to the WT and the Nbeal2-KO mice (Fig. 5D, magenta bar). The pool of DCX+ cells remained unaffected by thrombocytopenia (Fig. 5E). Next, we focused our analysis on the site of lesion in the CC (Fig. 5G–K). In WT mice, demyelination resulted in a significant increase in the total cell density at 7 dpl (Fig. 5G, black bar) and in the percentage of PCNA+ cells- at 5 dpl and 7 dpl (Fig. 5H), as expected due to the

recruitment of mitotically active macrophages and microglial cells. It also led to the significant decrease in the percentage of OLIG2+ cells at 5 dpl and 7 dpl (Fig. 5I), the cell lineage targeted by lysolecithin, as was also demonstrated by PLP staining (Fig. 4E–J). To assess more specifically the response of OPCs, we calculated the fraction of OLIG2+ cells that co-expressed PCNA and their presence was significantly increased after demyelination (Fig. 5J). Nbeal2-KO mice showed no differences to WT in all the above parameters (Fig. 5G–K, cyan bars). In contrast, short-term, severe, thrombocytopenia affected the cells of the oligodendroglial lineage in the CC, both at the uninjured and the injured hemispheres. The percentage of OLIG2+ cells was found to be increased compared to WT and Nbeal2-KO mice (Fig. 5I, magenta bar); even though the fraction of proliferating OLIG2+ cells did not show differences among the animal models (Fig. 5J). The presence of newly formed, maturing, oligodendrocytes was assessed by immunostaining for CNPase (Supplementary Fig. 4D–G and Fig. 5K). Their presence was significantly increased in all experimental groups at 7 dpl, compared to the uninjured CC, with no differences between the three groups (Fig. 5K).

This in vivo analysis generated two interesting observations. Firstly, that there was a clear difference in the response of the cells of the SEZ and of the CC between Nbeal2-KO mice and mice with chemical thrombocytopenia; with the former showing no differences from WT. Secondly, that transient, severe, thrombocytopenia resulted in increased numbers of OLIG2+ cells; hence, resembling and not contrasting the effects observed in the OPC:platelet co-culture assays. To investigate these two conflicting results, we revisited the tissue samples immunostained for CD41 and we observed that transient thrombocytopenia was accompanied by the enhanced presence of extravasated platelets, infiltrating the SEZ and the CC parenchyma (Fig. 5F, L), after the LPC-induced injury.

## Molecular signature of NSPCs co-cultured with platelets

Overall, our data revealed that when platelets are in proximity to NSPCs they can affect their cell behaviour. To gain insight on the molecular signature of the platelets' effects, we performed bulk RNAseq analysis in co-cultures, subculturing each biological NSPC sample (*n* = 3) in three conditions: 3 days in proliferation medium, 3 days in differentiation medium, and 3 days in differentiation medium in the presence of platelets at 1:75 ratio. This ratio was chosen because, apart from allowing good NSPC survival rates (Fig. 1E) and leading to significantly increased numbers of Sox2+ NSPCs (Fig. 1F), it was also relevant to the cell:platelet ratio at the sites of platelet injections.

To gain a more in-depth understanding of the response of cultured NSPCs to platelets at this density, we also looked at gene expression levels for *Ki67* and for the key oligodendroglial marker *Sox10*. Their expression was found to be at levels in between those seen in NSPCs cultured alone in proliferation and differentiation conditions (Supplementary Fig. 5A, B). Regarding neurogenesis, we complemented the counts of DCX+ cells (Supplementary Fig. 2B) by looking at the maturity of neuroblasts by calculating the co-expression of SOX2 (an indication of stemness and immaturity) and the number of cell processes. Both parameters were found to be unaffected by platelets (Supplementary Fig. 5C–F). Finally, prompted by the in vivo results, we tested if the effects of platelets required the direct

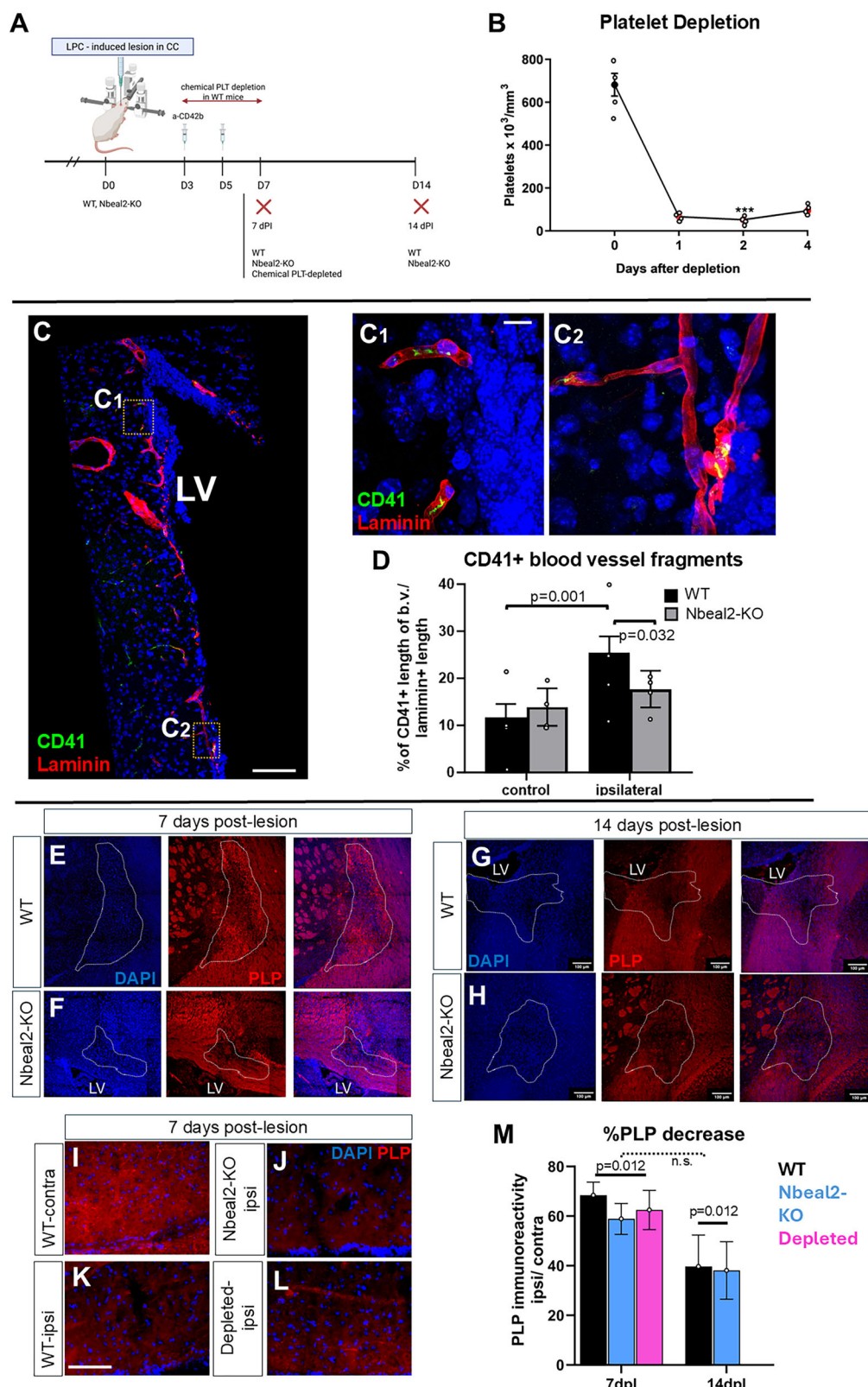

proximity with NSPCs by comparing three different culture conditions. NSPCs were cultured for 3 days in differentiation conditions, in the presence of: (a) platelets (1:75), (b) conditioned media obtained at the end of 3-day NSPC-platelet (1:75) co-cultures, and (c) conditioned media obtained at the end of 3-day NSPC-only cultures. This experiment showed that the exposure of NSPCs to medium conditioned by platelets had different effects from

the direct co-culture with platelets, as conditioned medium did not lead to an increase in the number of Sox2+ cells and exerted a toxic effect (Supplementary Fig. 5G, H).

The RNA-seq analysis confirmed the markedly different gene expression profile of NSPCs maintained in proliferation versus differentiation status (Fig. 6A), with 4047 genes (30% of the 13477 genes

**Fig. 4 | Long- and short-term thrombocytopenia in mice and the effects on demyelination in the corpus callosum. A** Schematic illustration of the demyelination experiment, with a lysolecithin (LPC) injection performed in the corpus callosum (CC) at day 0. Wild-type (WT), Nbeal2-KO mice, or mice with depleted platelets were killed at different days post-lesion (dpl). Acute thrombocytopenia was induced in wild-type mice via two i.p. injections of the CD42b antibody at 3- and 5-dpl. These mice were killed at 7 dpl. **B** Graph showing the number of platelets in blood samples taken from mice before and after the injection (on days 0 and 2) of the anti-CD42b antibody. **C** High magnification images of the SEZ of an Nbeal2-KO mouse at 7 dpl, after immunostaining for laminin (in red, to mark blood vessels) and CD41 (in green, to mark platelets). The areas outlined in (**C**) are presented in (**C1 & C2**), showing CD41+ platelets within blood vessels. **D** Graph showing the mean % of the length of blood vessels (immunopositive for laminin) covered by CD41+ immunopositive platelets, in the brain hemispheres ipsilateral (ipsi) and contralateral (contra) of the demyelinating lesion (n = 3 per experimental group). **E–H** Microphotographs of the CC, in WT and Nbeal2-KO mice, at the site of demyelination, at 7- and 14-dpl, after immunostaining for PLP. The area of lesion was determined based on cell density and is outlined with the white, dotted line. **I–L** Higher magnification microphotographs of the CC, in WT, Nbeal2-KO mice and platelet-depleted mice, at the site of demyelination and at the contralateral site, at 7 dpl, after immunostaining for PLP. **M** Graph showing the density of PLP immunoreactivity at the area of lesion, as a percentage of the contralateral, un-lesioned, site at 7- and 14-dpl (n = 4 per experimental group). [scale bars: 100 μm in (**C & E–H**); 10 μm in (**C'**); 50 μm in (**I**). In the graphs, bars show mean values and error bars the SEMs. Statistical analysis was performed using 1-way ANOVA, followed by the Tukey post-hoc analysis, in (**B**), with the three starts indicating a p < 0.001 for 1, 2, and 4 days post-injection. 2-way ANOVA, followed by the Tukey post-hoc analysis, was performed in (**D**), and the statistical differences are shown with horizontal brackets and the respective post-hoc values. 3-way ANOVA, followed by the Tukey post-hoc analysis, was performed in (**M**). The overall p-value for the effect of demyelination (versus the control hemisphere) is shown. No overall effect was found for experimental group (not shown), or for time-point (n.s. not significant). LV Lateral Ventricle].

expressed in the two conditions) exhibiting significantly altered levels of expression. The gene expression profile of NSPCs co-cultured with platelets remained closer to that of differentiating NSPCs (Fig. 6B, C). However, there were 271 genes with significantly upregulated and 235 genes with significantly downregulated expression (Fig. 6D, E) when NSPCs were co-cultured with platelets, compared to NSPC monocultures, all in differentiation conditions. A Gene Ontology (GO) analysis in the most upregulated genes revealed the activity of platelet-derived growth factor (PDGF) (Fig. 6F), confirming that the effects we monitored were induced by the presence of platelets. Notably, out of the 20 genes most significantly upregulated, at least nine (Lcn2, Steap4, Ccl2, Serpina3h, Npas1, Chil1, Egr2, and Cxcl1) have been previously shown to be involved in NSPC function, maintenance, or proliferation, indicating that platelets are able to affect directly the NSPC gene machinery. Iron metabolism, a key process in the regulation of proliferation and of the response to inflammation[51], mediated by Lcn2 and Steap4, as well as inflammatory-response genes, were also found to be upregulated in NSPCs by the presence of platelets. It should also be noted that the expression of the Cxcr3 gene, encoding for the receptor mediating the effects of platelet factor 4, which has been reported to act on NSPCs[36,37], was not found to be upregulated by platelets.

## Discussion

The activity of NSPCs is strongly dependent on the coexistence of supporting vasculature, as has been shown during brain development[52–54] and in the brain of songbirds, in areas with seasonal variation in neurogenesis[55]; while similar evidence exists for oligodendrogenesis[21,22,56]. In the postnatal mammalian brain, the SEZ niche vasculature is different from that of other periventricular areas, in being significantly denser[12,13], with lower blood flow[14], higher leakiness[15], and with NSPCs directly interacting with endothelial cells[16] and being affected by systemic factors[57,58].

Platelets, circulating in very high numbers within the vasculature, are now recognised to be regeneration-enhancing cells, based on: (a) their rich in growth factors cargo and their ability to interact with multiple cell types and the extracellular matrix[27–29,59], (b) their unlimited availability for autologous and heterologous use and, (c) the increasing evidence of safe and beneficial application in different systems (such as dental and bone regeneration)[32,33,60,61]. Human platelet lysate has been shown to support rat NSPC survival in vitro[39], and the intracerebroventricular injection of platelet-rich plasma promotes SEZ neurogenesis in mice[41]. Moreover, Platelet Factor-4, acting through CXCR3, was recently shown to increase the activity of, and to rejuvenate, NSPCs in the hippocampal niche[36,62] and to attenuate the effects of ageing to cognition[37,38]. Nevertheless, the full range of the effects of platelets on the brain's NSPCs, the molecular pathways that mediate this activity and the conditions under which platelets exert their actions remain, largely, elusive. This is due to the many parameters that can control and confound the effects of platelets on specific cell types. Of special basic and translational biology interest is to determine if circulating platelets can act on the brain's progenitor cells from within blood vessels, i.e., from their usual location, or if their extravasation is a necessary condition. To address these questions, we used a range of approaches: from co-culturing NSPCs and OPCs with platelets, to injecting platelets directly in the brain parenchyma and investigating the behavior of the brain's progenitors in mice with different levels of reduction in the numbers of circulating platelets.

A key parameter that we found influencing the effects of platelets is quantity. The in vitro data revealed an enhancement of OPC differentiation and maturation only at high platelet numbers (Fig. 7A). In the case of NSPC cultures, a dichotomous and gradient effect in the expression of Ki67 was observed. At low NSPC:platelet ratios the number of Ki67 immunopositive cells were significantly reduced, only to become significantly increased at higher ratios, with only increased levels of Ki67 mRNA at middle ratios. On the other hand, a significant increase in the percentage of SOX2+ cells was observed in all tested platelet densities (Fig. 7B). Recent experimental work has shown that the sustained presence of platelets in areas of demyelination diverts their effects from beneficial to detrimental[40]. In addition, elevated numbers of circulating platelets have been correlated with adverse effects of antipsychotic drugs in patients with schizophrenia[63]. Such observations highlight the complex activity of platelets, either those circulating or those having infiltrated the nervous tissue, and raise concerns, as their clinical use becomes wider and might be accompanied by a "the more, the merrier" approach.

Another crucial parameter that can determine the activity of platelets is the wide range of factors they carry, that enable them to act upon different cell types within the same location. Our in vitro approach, co-culturing platelets separately with SEZ NSPCs or OPCs, revealed the creation of a pro-stemness environment (enhanced expression of SOX2 and proliferation) for NSPCs and of a pro-differentiation environment for OPCs (Fig. 7A, B). The in vivo approaches confirmed both these results: the injection of platelets in the striatum led to the emergence of increased numbers of proliferating SOX2+ cells (Fig. 7F), while the presence of platelets in areas of demyelination, in severely thrombocytopenic mice, led to increased numbers of oligodendroglial lineage cells, without affecting the proliferation or the number of OPCs (Fig. 7C).

A third crucial parameter is the microenvironment in which platelets operate. Their presence in a pro-proliferation in vitro environment did not lead to significant effects on NSPCs, apart from a strong trend for increase in Olig2+ cells. In contrast, their presence in pro-differentiation conditions was sufficient to create a niche microenvironment, supporting SOX2 expression and mitotic activity. This was also confirmed in vivo, with NSPCs in the niche remaining largely unaffected by injected or infiltrating platelets, except for a strong increase in the presence of Olig2+ cells (Fig. 7D), but with dormant neural progenitors in the striatum being responsive (Fig. 7C, F).

Several observations revealed that the presence of platelets at the proximity of NSPCs and OPCs is of crucial importance for their effects to

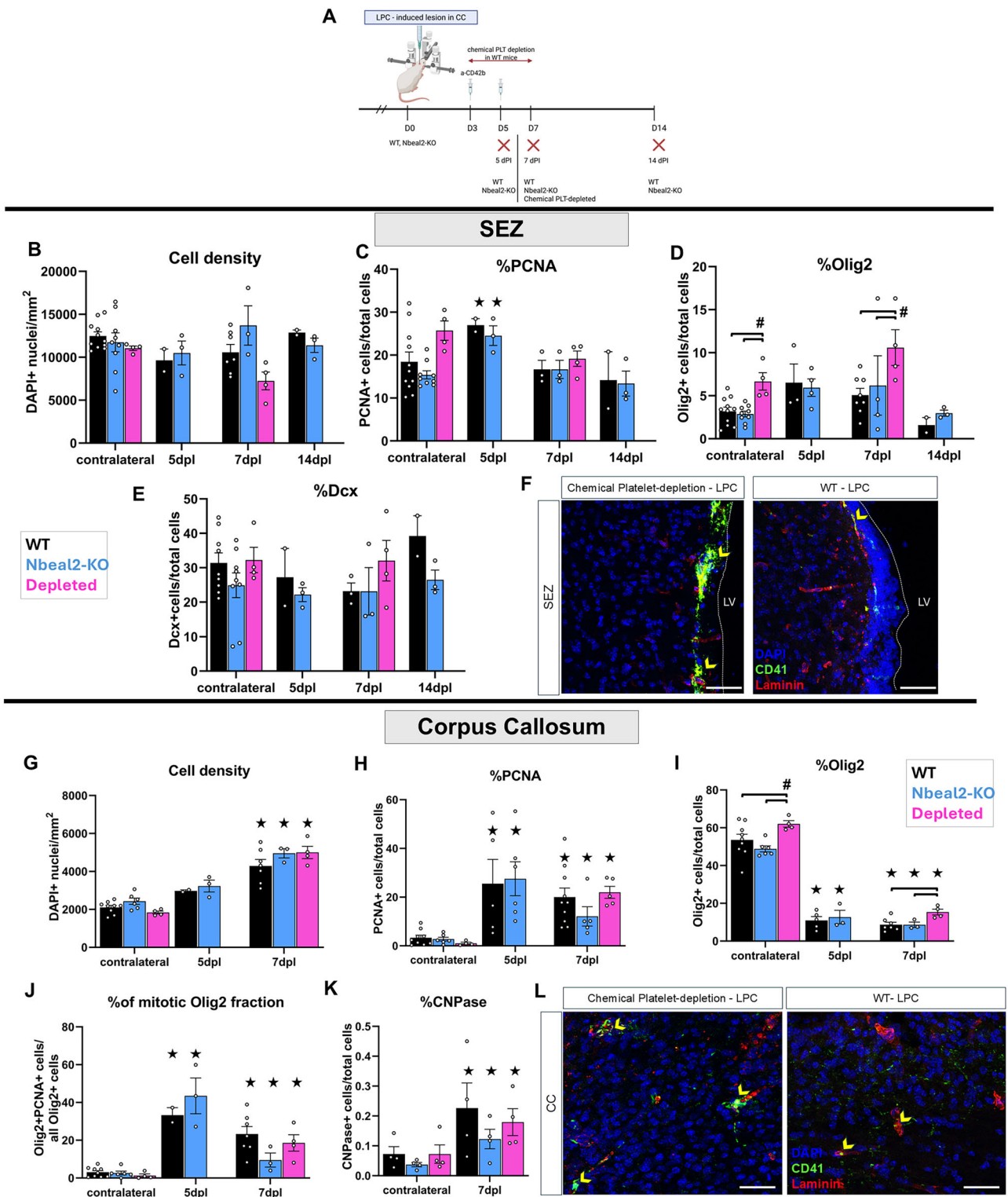

occur. NSPCs showed the same behavior (increased numbers of mitotic and SOX2+ cells) when co-cultured with platelets (Fig. 7B) and at the site of direct, intracerebral, injections of platelets in the striatum (Fig. 7F). The effects of platelets on oligodendroglial lineage cells were observed in co-cultures (Fig. 7A) and in mice with increased presence of parenchymal platelets. This "paradoxical" result was explained by the fact that very low numbers of circulating platelets led to the increased presence of platelets, infiltrating the SEZ and the CC parenchyma (Fig. 7C, D), suggesting the occurrence of micro-hemorrhages. In contrast, the reduction in the number

of circulating platelets down to a level that did not lead to such infiltration, in Nbeal2-KO mice, did not result in significant effects on NSPCs or OPCs, indicating that intravascular platelets do not act on the brain's progenitor populations. One possible explanation for this is that many platelet factors, with PDGF being a prime example[64], cannot cross the blood-brain barrier. This contradicts previous experimental work in which the transient, chemical, depletion of circulating platelets resulted in perturbed NSPC/OPC function in the hippocampal niche and in the demyelinated spinal cord[36,40] and highlights the diverse and complex nature of the activity of platelets and

**Fig. 5 | The effects of thrombocytopenia on NSPCs and OPCs. A** Schematic illustration of the demyelination experiment, with a lysolecithin (LPC) injection performed in the corpus callosum (CC) at day 0. Wild-type (WT), Nbeal2-KO mice, or mice with depleted platelets were killed at different days post-lesion (dpl). Acute thrombocytopenia was induced in wild-type mice via two i.p. injections of the CD42b antibody at 3- and 5-dpl. These mice were killed at 7 dpl. **B–E** Scatter plot, bar graphs showing the total cell density (**B**) and the percentage of PCNA+ (**C**), Olig2+ (**D**) and Dcx+ (**E**) cells, in the SEZ of WT (black bars), Nbeal2-KO (cyan bars) and mice with chemically depleted platelets (magenta bars) in the control (uninjured) brain hemisphere and in the injured hemisphere, at different time-points. **F** Microphotographs focusing on the SEZ, in mouse brain sections derived from the hemisphere of lysolecithin (LPC) demyelination, at 7 dpl, in WT mice and in WT mice injected twice with the anti CD42b antibody to induce severe thrombocytopenia. The sections were immunostained for laminin (in red, to label blood vessels) and CD41 (in green, to label platelets). Note the presence of platelet aggregates (indicated by yellow arrowheads) in the SEZ parenchyma only after chemical depletion. **G–K** Scatter plot, bar graphs showing the total cell density (**G**) and the percentage of PCNA+ (**H**), Olig2+ (**I**), of the mitotic fraction of Olig2+ cells

(**J**) cells, as well as of CNPase+ cells (**K**) in the CC of WT, Nbeal2-KO and of mice with chemically depleted platelets in the control (uninjured) brain hemisphere and in the injured hemisphere, at different time-points. **L** Microphotographs focusing on the CC, in mouse brain sections derived from the hemisphere of lysolecithin (LPC) demyelination, at 7 dpl, in WT mice and in WT mice injected twice with the anti CD42b antibody to induce severe thrombocytopenia. The sections were immunostained for laminin (in red) and CD41 (in green). Note, also here, the presence of platelet aggregates (indicated by yellow arrowheads) in the CC parenchyma only after chemical depletion. [scale bars: 50μm. The scatter plot bars show mean values and error bars the SEMs. The value of each experiment (biological sample) is depicted with a circle. 2-way ANOVAs were performed, followed by Tukey post-hoc analyses. The stars indicate statistically significant differences ($p < 0.05$) in the same experimental animal group when compared to the control (contralateral, uninjured) hemisphere. Significant differences ($p < 0.05$) within the same time-point but between different experimental groups are shown with connecting horizontal brackets and the hash marks. CC corpus callosum, LV lateral ventricle, SEZ Subependymal Zone, WT wild type].

the necessity for further, in-depth, investigation in multiple experimental and clinical conditions. Our data, showing that platelet-conditioned medium exerts different effects as compared to the direct platelet:NSPC contact (Supplementary Fig. 3G, H) also highlight that the activity of platelets is not mediated only by diffusible factors, but is also dependent on cell:cell, mechanotransduction, mechanisms, because platelets can sense and respond to changes in their microenvironment[65], possibly via integrins[30,59,66].

In terms of the molecular signalling that mediates the activity of platelets on NSPCs and OPCs, the role of PDGF as an OPC pro-differentiation factor has been well documented[67–69] and could explain the selective effect of platelets on cells of the oligodendroglial lineage, both in the SEZ and in areas of demyelination. The GO analysis showed that the PDGF pathway was active in NSPCs co-cultured with platelets, with at least two of the genes with upregulated expression (CCL2 and Chitinase-like protein 3) known to be induced by this growth factor[70–72]. At the same time, platelets led to increased expression of other genes that have been shown to control NSPCs. Lcn2[73,74], Spp1 (also known as osteopontin)[75–78], CLC2 (also known as MCP-1)[79], Serpina3[80], and Chitinase-like protein 3[81] have all been shown to enhance NSPC proliferation, an effect we observed in vitro. The formation of a pro-stemness environment by platelets could be mediated by CCL2, which induces the expression of pluripotency genes[82] and has been shown to act directly on NSPCs[83,84]. Similarly, Chitinase-like protein 3 is a known component of the neurovascular niche[81] and Serpina3 is expressed by reprogrammed parenchymal astrocytes[25]. Importantly, Lipocalin-2, Steap-4, and Spp1 (the 2nd, 3rd, and 4th most-highly expressed genes in NSPCs co-cultured with platelets), as well as Serpina3, are all pro-inflammatory molecules[75,85–88] that have been linked to neurodegenerative diseases, such as multiple sclerosis[87,89,90], while Lipocalin2 and Steap4 are also involved in the all-important regulation of iron metabolism. The expression of such pleiotropic genes could explain the dichotomous effects of platelets on NSPCs and OPCs, with positive and detrimental results dependent on levels and combination of expression, as well as on additional corroborating factors, such as the tissue microenvironment and the involvement of other cell types.

What becomes clear from this study is that circulating platelets are not a canonical regulatory element for NSPCs within their niche, or of OPCs in the CC. However, they can modulate the brain's progenitors when infiltrating the brain parenchyma, as can happen in areas of blood vessel damage, due to acute or chronic injury or inflammation. In such cases, platelets become modulators of NSPCs and OPCs, with possible pleiotropic activity which depends on their numbers and the tissue microenvironment. Our data revealed that the cells of the oligodendroglial lineage are especially responsive to platelets. Their differentiation was enhanced by the presence of platelets in vitro, and the total number of Olig2+ cells remained significantly increased after demyelination, in the presence of platelets. Our further analysis using different markers of the lineage, such as PLP and

CNPase, and the co-staining for Olig2 and PCNA, did not allow the identification of the specific maturation stage that is directly affected by platelets and this needs to be addressed in the future. This activity of platelets could be a novel target as part of neuroprotective and neuroregenerative strategies, especially because numbers of circulating platelets can be manipulated pharmacologically, as well as with plateletpheresis (removal of platelets) or platelet transfusion from donors. Moreover, platelets can be a valuable cell source for the improvement of NSPC/OPC-based transplantation strategies, either by priming cells before grafting or in co-transplantations. This option is facilitated by the easiness of collecting platelets for autologous use since platelet donations can be safely performed as often as every 2 weeks.

## Materials and methods
A more detailed version of the methods and the materials (e.g., catalogue numbers, list of antibodies, details of primers and of protocols) can be found in the Supplementary material.

Schematic illustrations found in Figs. 1–5 and in Supplementary Fig. 3 were made using Biorender. Figure 7 was generated using Adobe Illustrator (version 29.7).

### Animals
Male and female mice of the Bl6CBAC or of the 129sv background, between 2 and 4 months of age, were used in co-cultures, platelet depletion experiments and in the administration of BNN-20. For long-term thrombocytopenia, male mice of the same age, belonging to the Nbeal2 line, were used[50]. Male Sprague Dawley rats (9 weeks old; 225–280 × $g$) were used for the MCAo experiment[91] ($n = 3$) and for neuraminidase administration[48] ($n = 5$). Mice were maintained in steady light/dark cycle (12/12 h) with free access to food and water. Animal breeding, maintenance and handling was performed in accordance with the European Communities Council Directive Guidelines (86/609/EEC) for the care and use of Laboratory animals as implemented in Greece by the Presidential Decree 56/2013 and approved and scrutinized by the local Prefectural Animal Care and Use Committee in Patras (Protocol number: 118188/432/21-05-2020) and in Crete (Protocol number: 106357/31-05-2021).

### Isolation of platelets
Platelets were isolated from the vena cava inferna[92], by collecting approximately 1 ml of whole blood in tubes containing EDTA buffer (0.5 M, pH 8) as anticoagulant and subsequently separating and washing the platelet rich-plasma fraction (PRP) using Tyrode's buffer. Apyrase and PGE1 were added to the PRP in order to avoid platelet activation and PRP was then centrifuged. At the end, platelets were resuspended in NSPC or OPC culture medium. For labelling with the lipophilic dye PKH26 (PKH26GL, Sigma-Aldrich), platelets were processed according to the manufacturer's protocol, in the presence of PGE1. To assess activation, 50 μL of washed platelets were

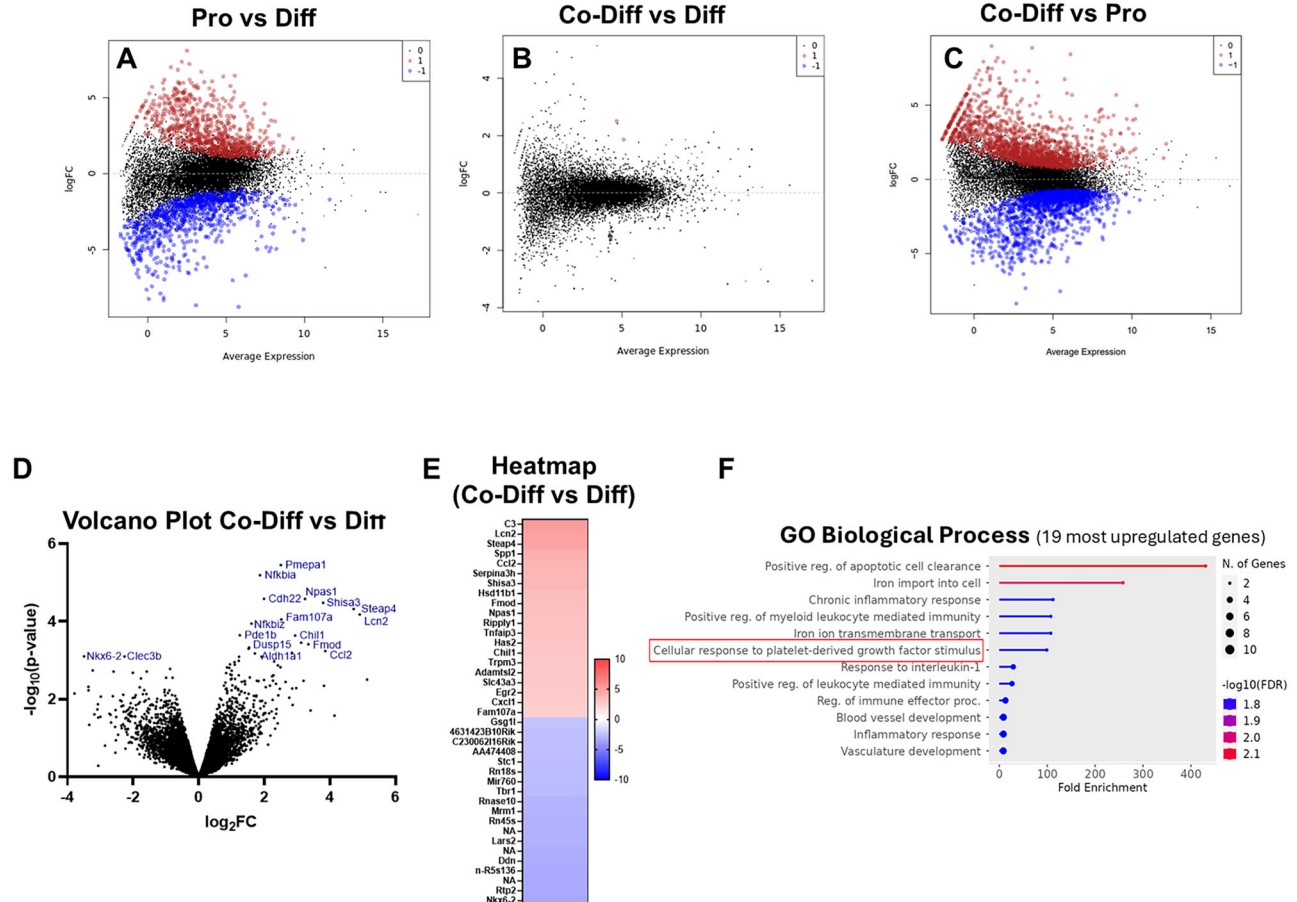

**Fig. 6 | Bulk RNA sequencing in NSPC/platelets co-cultures. A–C** Scatterplot matrices showing the differential expression of genes (those upregulated in red, those downregulated in blue and those with no change in black), when comparing pure NSPCs cultured in proliferation (Pro) and differentiation (Diff) conditions, as well as co-cultured in differentiation conditions with platelets (Co-Diff), in a NSPC:PLT ratio of 1:75. Note the marked differences found between the Pro and Diff (**A**), as well as between the Co-Diff and Pro (**C**) conditions. Also, note the similarity between the Co-Diff and Diff conditions (**B**). **D** Volcano plot of differential gene expression between NSPCs cultured alone in differentiation conditions and NSPCs co-cultured with platelets, in differentiation conditions at a 1:75 ratio. **E** Heatmap of the genes with the highest change of expression (upregulated in red, downregulated in blue) between the Co-Diff and Diff conditions. **F** Gene Ontology (GO) analysis of the biological processes in which are involved the 19 most upregulated genes, when comparing the Co-Diff and Diff conditions.

incubated with APC-labeled anti-CD41 (BioLegend) and FITC-labeled anti-CD62P (BD Pharmingen) (or a FITC-labeled IgG1λ isotype control, BD Pharmingen) and were analysed with the BD Accuri TM C6 Plus (BD Biosciences).

## Isolation and culture of NSPCs and of OPCs; co-cultures

Mouse brain NSPCs were isolated by dissecting and dissociating the SEZs as done previously[42] and were grown as neurospheres in the presence of FGF2 and EGF (Peprotech) at 20 ng/ml, 2% B27, and 1% N2 Supplementary (ThermoFisher Scientific). Neurospheres were passaged every 5 to 7 days, and those between passages 5 and 10 were used.

Primary OPC cultures were obtained from postnatal day 2 mouse cortices[93] after removal of microglial cells via orbital shaking (200 rpm, 1 h, 37 °C) and separation from astrocytes by vigorous shaking (16 h, 240 rpm, 37 °C). After 3 days in proliferation medium, cells were changed to differentiation medium (replacing PDGF-AA and hFGF-2 with 40 ng/ml T3).

Co-cultures were performed in multi-well plates, with cells grown on PDL-coated glass coverslips. NSPCs or OPCs were plated the day before the addition of platelets. A range of NSPC or OPC:platelet ratios were used (1:40, 1:75, 1:400, 1:1200, 1:3000) with the numbers of NSPCs/OPCs and platelets calculated with the use of a haematocytometer. Cells were co-cultured for 3 (NSPCs) or 4 (OPCs) days, without addition or change of medium. Conditioned medium was collected after 3-day cultures, in differentiation conditions, either of NSPCs alone, or of NSPCs and platelets at a 1:75 ratio. The medium was collected and rendered cell-free through centrifugation; it was used immediately.

## Experimental injury models and intracerebral injection of platelets

For demyelination, 1 µL of lysolecithin (1% w/v LPC in sterile PBS, L1381 or L4129, Merck Sigma-Aldrich) was injected, through a burr hole, in the CC at a flow of 0.1 µL/min. The stereotactic co-ordinates from bregma were: anteroposterior axis (AP) + 1.0 mm, mediolateral axis (ML) −1.0 mm, dorsoventral axis (DV) −2.2 mm from the dura. Mice were killed at 5, 7, or 14 dpi and the contralateral hemisphere was used as internal control. The Middle Cerebral Artery Occlusion procedure has been published elsewhere[91]. Briefly, right middle cerebral artery occlusion (MCAo) was performed under anaesthesia (n = 3) using the intraluminal filament technique. The animals remained anaesthetized for 60 min when the monofilament was withdrawn to allow for reperfusion and recovery; brain tissue was collected 4 weeks post MCAo. Administration of neuraminidase has been described in detail previously[48]. Briefly, 500 mU of *Clostridium perfringens* neuraminidase (N2876, Merck Sigma), along with 1 µg integrin-β1 blocking antibody and 0.5 µg Fibroblast Growth Factor 2 (in a total 1 µl volume), was bilaterally injected at a rate of 1 µl/min at (AP) 0.3 mm, (ML) ± 2.0 mm, (DV) 3.5 mm. Rats were killed at 7 days post-injection.

**Fig. 7 | Summary of in vitro and in vivo experimental results. A** The co-culture of OPCs with platelets leads to increased levels of differentiation towards myelin-forming oligodendrocytes. **B** The co-culture of NSPCs with different densities of platelets leads to a dichotomous effect on the expression of Ki67, with a significant reduction in Ki67+ cells at low platelet densities, an increase in *Ki67* mRNA at middle densities, and an increase in Ki67+ cells at high densities. Another effect is the increased presence of Sox2+ cells, across all tested densities of platelets. **C, D** A demyelinating lesion (area in green characterized by loss of oligodendrocytes) was induced in the CC proximal to the SEZ (area in yellow) by injecting lysolecithin, and the effects on OPCs (**C**) and NSPCs (**D**) were investigated. The reduction in the numbers of circulating platelets, in Nbeal2-KO mice, did not result in any changes compared to wild-type (WT) mice, both in the CC and the SEZ. The more severe, chemical, depletion of platelets led to the extravasation of platelets in the brain parenchyma, possibly due to micro-hemorrhages. This was accompanied by a significant increase in the presence of oligodendrocytes in the CC and of OPCs in the SEZ. **E** The increased accumulation of platelets within the vasculature of the SEZ, observed in response to neuraminidase-induced damage of ependymal cells, did not lead to changes in NSPCs. **F** The intracerebral injection of platelets in the striatum led to the emergence of mitotic Sox2+ cells. [The illustration is not in scale and only the necessary cell types are depicted].

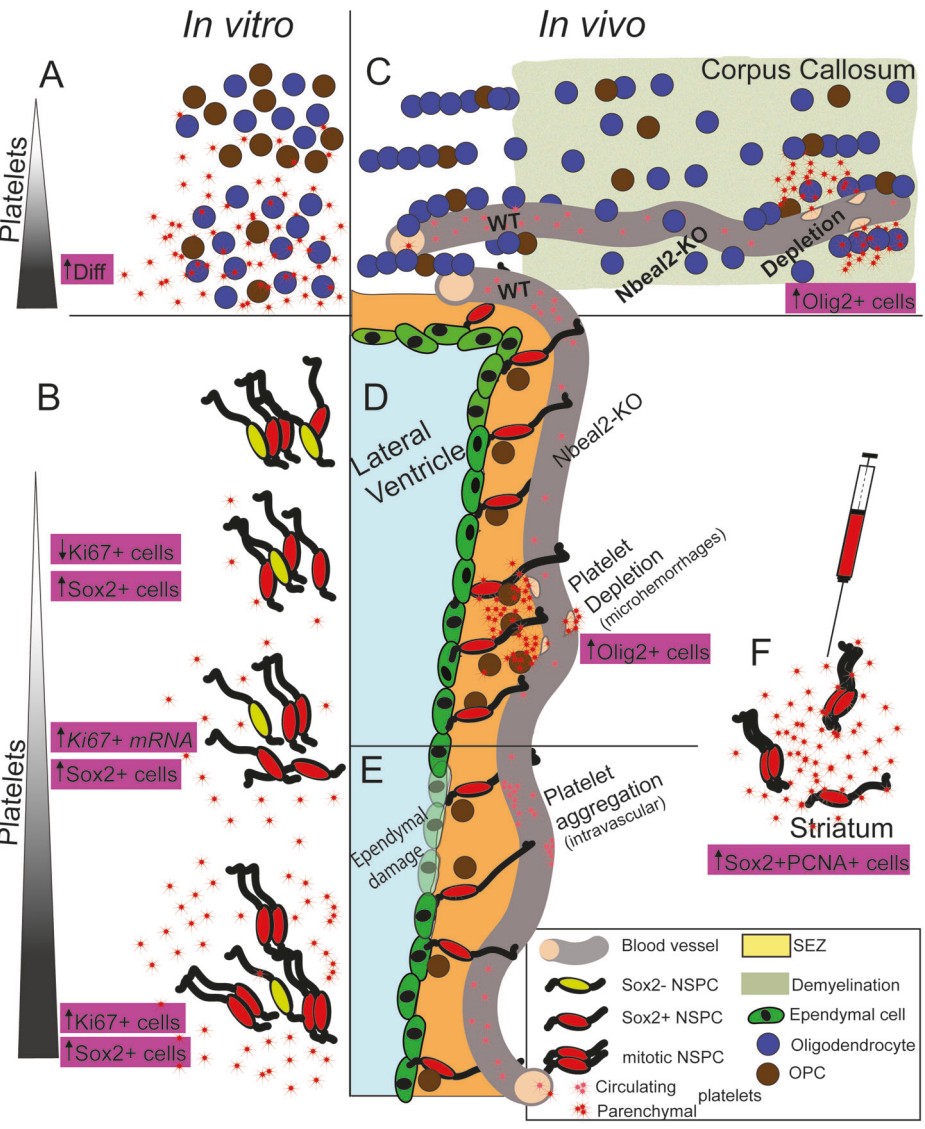

Administration of the microneurotrophin BNN-20 has been described before[49]. Briefly, mice received daily i.p. administration of BNN-20 (1 mg/ml in 1% ethanol, 0,9% NaCl; 100 mg BNN-20 per kg b.w.) (Bionature E.A. Ltd, Nicosia, Cyprus) or vehicle, during postnatal days 14–40.

For the stereotaxic injection of PKH26-labelled platelets, 30000 platelets (in 1 µL of DMEM or DMEM alone) were injected at a rate of 1 µL/min, with the Hamilton needle left in place for another 5 min. The coordinates were: (a) for the dorsoventral horn of the SEZ (AP) + 0.5 mm, (ML) ± 1.2 mm, (DV) −2.0 mm from the dura. (b) For the striatum (AP) + 0.1 mm, (ML) ± 1.8 mm, (DV) −2.3 mm from the dura.

**Transient thrombocytopenia**

For the chemical depletion of platelets, mice were injected twice, i.p., with anti-mouse CD42b antibody (0.6 µg/g of body weight, diluted in sterile PBS; #R300, Emfret Analytics, GMBH) or vehicle (sterile PBS). The first injection was performed three days after the induction of demyelination and it was repeated after 48 h (at 3 dpl and 5 dpl). Numbers of circulating platelets were monitored before the day of antibody injection, and subsequently at 1-, 2-, and 4-days post-injection. Approximately 80 µL of blood were collected from a cut at the tip of the tail, into Microvettes (SARSTEDT CB 300 KK2EE), and platelets were counted with an automated veterinary blood analyser.

**RNA-seq**

Whole transcriptome sequencing analysis was performed in NSPCs, cultured on PDL-coated wells of 6-well plates. Three independent cell cultures were performed (sample 1 derived from a 2-month-old male mouse; sample 2 derived from a 3-month-old female mouse; sample 3 derived from a 2.5-month-old male mouse) with 300000 cells plated and kept overnight in proliferation medium to achieve good adherence. The next day one well (per sample) was kept in proliferation conditions, in a second well, the medium was changed to differentiation medium and in the third, 22.5 million platelets (ratio 1:75) were added in differentiation medium. The cultures were kept for three days, without any media changes or additions. Total RNA extraction was performed with the NucleoSpin RNA Plus or the NucleoSpin RNA Plus XS (MACHEREY-NAGEL) kits. The ribosomal RNA was depleted using the RiboMinusTM Eukaryote Kit v2 (ThermoFisher Scientific Inc.) and quantification of the RNAs was performed using the Qubit RNA HS (High Sensitivity) Assay Kit (ThermoFisher Scientific Inc.) with the Qubit Fluorometer. rRNA-depleted total RNA was used for the preparation of cDNA libraries, with the Ion Total RNA-Seq v2 Kit (Thermo-Fisher Scientific Inc.). A unique barcode was added in each fragmented library using the Ion Xpress™ RNA-Seq Barcode 1–16 Kit (ThermoFisher Scientific Inc.). Quantitation of libraries was performed with the Qubit 1× dsDNA HS Assay Kit (ThermoFisher Scientific Inc.) and quality was

assessed using the Agilent High Sensitivity DNA Kit (Agilent Technologies) on the 2100 Bioanalyzer system. Template preparation and Ion 540™ chip (ThermoFisher Scientific Inc.) loading were performed on the Ion Chef system with the Ion 540™ Chef kit (ThermoFisher Scientific Inc.). The single-end sequencing was carried out on the Ion GeneStudio S5 system.

**Analysis of RNA-seq data.** The analysis of the sequencing data was performed on the Galaxy web platform through the usegalaxy.org public server[94]. Reads were mapped to mm10 with the STAR aligner (v2.7.8a) with default parameters except the following (–sjdbOverhang 75 –outSAMmapqUnique 255 –chimSegmentMin 18 –chimScoreMin 12 –outFilterType BySJout)[95]. The unmapped reads were aligned to mm10 using bowtie2 (v2.4.5) using the –very-sensitive-local preset[96]. The generated BAM files were then merged using Picard MergeSamFiles (v2.18.2.1), followed by filtering out the unmapped reads with the samtools view (v1.15.1)[97]. Gene expression was counted with featureCounts (v2.0.1) and the differential expression analysis was performed with limma-voom (v3.50.1), with genes with CPM > 0.5 in at least one sample only considered[98,99]. All raw and processed sequencing data are available at the GEO Repository under the Accession Number GSE256325. Further functional enrichment analysis of the differentially expressed genes was performed on the PANTHER platform[100].

### RT-PCR
Cell lysis and RNA isolation was performed according to the manufacturer protocols using the Kit NucleoSpin RNA Plus kit for NSPCs in proliferation conditions and the NucleoSpin RNA Plus XS kit (MACHEREYMACHEREY-NAGELNAGEL) for NSPCs in differentiation conditions. cDNA was generated using the High-Capacity cDNA Reverse Transcription Kit (Applied Biosystems). Real-time PCR was performed using the KAPA SYBR FAST Universal qPCR Kit (KAPA Biosystems) with actin-b and GAPDH used as housekeeping.

### Immunohistochemistry and Immunocytochemistry
Mice were transcardially perfused, forebrains were post-fixed in 4% PFA and frozen at −80 ℃; immunostainings were performed on 14-μm-thick coronal sections. Cells were fixed in 2% PFA. Immunofluorescence was performed using standard protocols[42]. Images were obtained with confocal microscopy (Leica microsystems, TCS SP8) with the 40× objective lens and were processed for cell counts using Fiji (https://imagej.net/Fiji) or LasX (Leica). Cell counts were performed blind, from at least 2 sections and at least 4 optical fields per section or in -at least- 10 optical fields per coverslip. For PLP measurements, we used densitometric analysis on stacks of 10 layers (2 μm-wide each). The area of lesion was defined by the accumulation of DAPI+ nuclei. The fraction of CD41+ blood vessels was calculated by manual measurements of laminin+ and CD41+ vessel fragments using Fiji.

### Statistics
Statistical analyses were performed using Excel (Microsoft® Excel® for Microsoft 365 MSO, Version 2409), SPSS (IBM), or GraphPad Prism 10 software. Depending on the comparison, and as stated in the relevant parts of the text, we performed 3-way, 2-way, or 1-way ANOVAs, and student's *t*-test (paired or unpaired).

### Ethical approval
Research reported here included local researchers throughout the research process—study design, study implementation, data ownership, intellectual property, and authorship. Roles and responsibilities were agreed amongst collaborators, whenever possible ahead of the research and for several researchers, their involvement was used as a capacity-building plan. The study has been approved by local ethics review committees, in terms of animal use, so as to maintain the higher possible standards.

### Reporting summary
Further information on research design is available in the Nature Portfolio Reporting Summary linked to this article.

## Data availability
All the data supporting the graphs are available in Supplementary Data 1 and 2. All the raw data (numeric or images) supporting the findings of this study are not openly available due to reasons of sensitivity and due to size limitations, but can be made available upon reasonable request to the corresponding author. RNA-sequencing raw data are available from the Gene Expression Omnibus under accession code GSE256325.

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

## Acknowledgements

The work was funded by a research grant from the Hellenic Foundation for Research and Innovation (HFRI-FM17-3395) to I.K. F.J.R. is funded by FONDECYT Program Regular Grant Number 1201706 from Agencia Nacional de Investigación y Desarrollo (ANID, Chile), the Sigrid Jusélius Foundation, and the Finnish MS foundation (Suomen MS-Säätiö). In addition, thanks to PROFI 6N° 336,234 of the Research Council of Finland. M.A. received a PhD Scholarship from the Bodossaki Foundation [46th regular Scholarships Programme] and she was also supported by Greece and the European Union (European Social Fund-ESF) through the Operational Programme «Human Resources Development, Education and Lifelong Learning» in the context of the Act "Enhancing Human Resources Research Potential by undertaking a Doctoral Research" Sub-action 2: IKY Scholarship Programme for PhD candidates in the Greek Universities. The work was also supported by Erasmus+ traineeships [Erasmus+ Mobility Programs 2016–2017 and 2021–2022] to C.D. and M.A., respectively. S.K. was a recipient of a 2022 Sidney Altman graduate fellowship by Fondation Sante. The support of Professor Ludwig Aigner, Ms Heike Mrowetz, and of Dr. Diana M Bessa de Sousa (Institute of Molecular Regenerative Medicine, Paracelsus Medical University, Salzburg, Austria) is gratefully acknowledged. We are grateful to Dr Ioanna Sandvig and to Professor Axel Sandvig

(Norwegian University of Science and Technology, Trondheim) for the provision of brain tissue of Sprague-Dawley rats after MCAO.

## Author contributions

The contribution of each co-author is described below: C.D., M.A., S.K.: data curation, formal analysis, methodology, validation, visualization, writing–review & editing. D.D., D.L., A.K., A.R.P.: data curation, formal analysis, validation, visualization, writing–review & editing. T.M., A.D., F.K., M.N., K.M., K.K., E.A., E.T.: data curation, formal analysis, validation. C.S., D.K., R.J.M.F., C.G., F.J.R.: data curation, methodology, project administration, resources, supervision, writing–review & editing. I.K.: conceptualization, data curation, formal analysis, methodology, project administration, resources, supervision, validation, visualization, writing–original draft, review & editing.

## Competing interests

The authors declare no competing interests.

## Additional information

[1]Laboratory of Developmental Biology, Department of Biology, University of Patras, Patras, Greece. [2]Department of Basic Science, University of Crete Medical School and Institute of Molecular Biology and Biotechnology, Foundation for Research and Technology Hellas, Heraklion, Greece. [3]School of Life Sciences, University of Westminster, London, UK. [4]Laboratory of Stem Cells and Neuroregeneration, Institute of Anatomy, Histology and Pathology, Faculty of Medicine, Universidad Austral de Chile, Valdivia, Chile & Center for Interdisciplinary Studies on the Nervous System (CISNe), Universidad Austral de Chile, Valdivia, Chile. [5]Department of Biochemistry, School of Medicine, University of Patras, Patras, Greece. [6]Cambridge Stem Cell Institute & Department of Clinical Neurosciences, University of Cambridge, Cambridge, UK. [7]Altos Labs, Cambridge Institute of Science, Cambridge, UK. [8]Cambridge Stem Cell Institute & Department of Haematology and NHS Blood and Transplant, University of Cambridge, Cambridge, UK. [9]Translational Regenerative Neurobiology Group (TReN), Molecular and Integrative Biosciences Research Program (MIBS), Faculty of Biological and Environmental Sciences, University of Helsinki, Helsinki, Finland. [10]Present address: MRC Centre for Neurodevelopmental Disorders, Institute of Psychiatry, Psychology and Neuroscience, King's College London, London, UK. [11]Present address: Centre for Developmental Neurobiology, Institute of Psychiatry, Psychology and Neuroscience, King's College London, London, UK. [12]Present address: Laboratory of Brain Exosomes and Pathology – ExoBrain, Institute of Biosciences and Applications, National Centre for Scientific Research (NCSR) "Demokritos", Athens, Greece. [13]These authors contributed equally: Christina Dimitriou, Maria Anesti. ✉e-mail: i.kazanis@westminster.ac.uk

