## [Transparent Peer Review file · Communications Biology]

Platelets regulate neural and oligodendroglial progenitors when infiltrating the brain parenchyma

Corresponding Author: Dr Ilias Kazanis

Version 0:

Reviewer comments:

Reviewer #1

(Remarks to the Author)

In this paper, the authors investigated the regulatory role of platelets for neural stem/progenitor cells (NSPCs) and oligodendrocyte progenitor cells (OPCs) both in vitro and in vivo, and they showed that platelets created a pro-stemness environment for NSPCs and enhanced the differentiation of OPCs.

This paper deserves publication in communications biology, if the authors properly address the following points.

1. In figure 1E, the authors showed the effect of NSC-PLT ratio on the changes in % of SOX2+ cells normalized per "no Platelets". They should show the percentage of SOX2+ cells in their NSPC preparation in the absence of platelets.
2. To my eyes, the shape of the blood vessel in the inset (C) of Figure 2H' is different from that in Figure 2H-C.
3. I would like to know what kind of cells were SOX2-/PCNA+ in the striatum injected with platelets.
4. Why didn't they detect any difference in the fraction of blood vessels with adherent CD41+ platelets between WT and Nbeal2-KO mice in non-injured hemisphere?
5. They examined the effect of lysolecithin on WT mice, Nbeal2-KO mice, and chemically-induced thrombocytopenic mice, but they showed only the results in WT and Nbeal2-KO mice. I would like to know the results in chemically-induced thrombocytopenic mice.
6. "Do" in Figure 4A should be "D0".
7. In line 307, they wrote "SOX2 is also expressed by ependymal cells and reactive astrocytes". They should show the evidence supporting this.
8. In line 310, isn't (Fig5F-G) proper, instead of (Fig5D-G)?
9. In line 327, "form" should be "from".
10. Their RNA-seq analysis revealed the activity of PDGF on NSPCs cocultured with platelets. PDGF is generally considered to be released from activated platelets. They should show the percentage of activated (CD62P+) platelets in their NSPC-platelet co-cultures. Conditioned media obtained at the end of 3-day NSPC-platelet (1:75) co-cultures did not lead to an increase in the number of SOX2+ cells. They should explain this.

Reviewer #2

(Remarks to the Author)

The manuscript titled "Platelets regulate neural and oligodendroglial progenitors when infiltrating the brain parenchyma" by Dimitriou et al. explores the role of platelets in modulating the behavior of neural stem/progenitor cells (NSPCs) and oligodendrocyte progenitor cells (OPCs) within the brain. The conclusions from this study emphasize the supportive and complex roles platelets can play in neuro-regenerative strategies, particularly in conditions where platelets infiltrate the brain parenchyma due to vascular damage or pathological conditions. The findings are interesting and suggest significant translational aspects. However, certain areas need clarification and improvement to enhance the scientific rigor, presentation, and broader relevance. Special attention should be given to the quality and readability of the figures as outlined below.

General comments:

1. Unless justified, I recommend to use consistent terminology throughout the manuscript. For instance, 'NSPC' and 'NSC' are used interchangeably. Stick to one term (e.g., "NSPC") and ensure it is applied consistently (e.g., Fig 1, Fig S3).
2. I suggest to directly label specific figures or panels within figures to explicitly indicate the experimental model that has

been used to generate the data. This will greatly help readers to follow the experimental design and the corresponding results.

3. A more detailed explanation of how platelets influence NSPC and OPC behaviors could be proposed. For instance incorporating a pathway or model diagram would enhance understanding and the scientific impact of the manuscript.

4. The controls used, especially during in vivo experiments, should be more clearly presented to reinforce the strength of the conclusions.

5. The manuscript has a few typos that should be corrected: "neuramidase", pertubated - perturbed (discussion); "in re" (Fig 1 legend); "exeperimental" (fig 5 legend); "form" --- > "from" (Methods"; CD1+ ---- > CD41+ (Fig 2H).

6. « This work advances the potential to use this naturally unlimited, and suitable for autologous use, source of cells in regenerative biomedicine » in the introduction and b) "their unlimited availability for autologous and heterologous use" in the discussion are statements that deserve further exploration and discussion with some brief explanations on how this can be achieved with a reference to current developments seen in the use of allogenic platelets materials from healthy donors for brain administration and treatment of CNS disorders.

Specific comments:

M&M: (a) Platelet isolation: what kind of anticoagulant was used? What were the volumes processed? Platelets are not "redissolved" but "resuspended".

Figure legends: Please ensure that all statistical methods are clearly described in the legends and present results with consistent and appropriate indicators of statistical significance (e.g., p-values, error bars). Also please specify how many independent experiments were conducted. N (independent experiments) = ?

Fig 1 and others: It is difficult if not impossible to see all data points and in some instances the the number of data points seems insufficient and not able to support the calculated averages. In addition it is stated that "The value of each experiment (biological sample) is depicted with a circle and is normalized per the "no-platelets" culture of the same sample, which is at "1.0" and is indicated by the red, dotted line": why the authors does not show the individual data points for the "no-platelets" condition in addition to the dotted line for transparency of the data?

Fig 1D-E: Please clarify whether the increased Sox2 expression without DAPI changes indicates increased Sox2 protein levels without changes in cell number or nuclear size. Specify whether this signifies "stemness" and describe how Sox2+ cells were quantified (e.g., area- or cell count-based). One condition has a large error bar: is there any experimental explanation.

Fig 1F: Clarify the discrepancy between the 3.5-fold Ki67 increase and the unchanged DAPI staining. This requires further clarity.

Fig 3F-L: the figure organization is confusing and it is recommended to reorganize the panels for improved readability by grouping images with similar experimental conditions (e.g., group striatum/SEZ sites and marker series).

Fig 3H, include a control image with the same frame selection as the injection site and provide high-magnification images of both conditions to enhance clarity.

Fig 4D: please define "b.v." to avoid confusion.

Fig 5A-C To enhance the clarity of the results, a representative image of the dysmyelinated hemisphere in WT mice for direct comparison with control hemisphere should be included.

Fig 5F Please add statistical significance indicators.

Fig S1E, F: Can the resolution of the images be improved ? Clarify the presence of erythrocytes in Fig S1E if applicable.

Reviewer #3

(Remarks to the Author)

In this paper by Dimitriou et al., the authors studied the effects of platelets on neural and oligodendroglial progenitors both in vitro and in vivo. The authors suggested that when co-cultured with NSPCs or OPCs, platelets were able to create a pro-stemness microenvironment and promote OPC differentiation. In vivo, thrombocytopenia without platelet tissue infiltration in Nbeal2 knockout mice did not affect the brain's progenitors. However, when platelets were injected directly into the striatal parenchyma or extravasated in severely thrombocytopenic mice, there was an increase in Olig2+ and Sox2+ cells.

The manuscript addresses a potentially interesting topic, however, the presentation of the data is not completely clear, and the narrative structure could be significantly improved. I recommend reviewing the text to create a more logical flow and providing additional clarification or context for the presented data.

Major points:

1) It is not clear how the proportion of NSPCs/OPCs to platelets was calculated and chosen. Does the ratio of "NSPC:platelets" 1:1000 mean that platelets are 1000 times more abundant than the other cells? In a previous paper published by the same group (Philp et al., eLife 2023;12:RP91757. DOI: [https://doi.org/10.7554/eLife.91757] (https://doi.org/10.7554/eLife.91757)), OPCs were subjected to significantly lower concentrations of platelets (1%, 5%, and 10%). Please clarify.

2) In Figure 2, a quantification of the observations would enhance the significance of the data.

3) I have some concerns regarding how the data on the effects of platelets on OPC differentiation have been presented. Firstly, the authors stated: "the numbers of CC1+ and MBP+ cells were also found to be significantly increased in the presence of 3000 platelets per OPC (Fig. 1o)" [line 127]. However, the graph in Fig. 1o only shows CC1+ and CC1-MBP+ cells, and it is unclear whether these are expressed as percentages or absolute numbers (as indicated in the figure legend, line 165: "Graphs showing [...] the number of CC1+ and MBP+ cells per optical field"). To improve clarity and consistency, please modify this graph to present the data as:

- The percentage of CC1+ and MBP+ cells over the total number of cells.
- The percentage of CC1+ - MBP+ double-positive cells over the total number of CC1+ cells.

- 4) How was the number of stereotactically injected platelets determined? Is the ratio between cells and platelets similar to that used in the in vitro experiments? Additionally, the effects on the oligodendrocyte population should be investigated.
- 5) The in vivo results regarding the effects of long-term and short-term thrombocytopenia are unclear.
- 5a) Please include representative images in Figure 5 for at least the condition where statistical significance is observed, highlighting the area where cells were counted.
- 5b) Results should be expressed as the number of positive cells per mm² (n° positive cells/mm²).
- 5c) The effects on oligodendrocytes are poorly characterized. At the very least, a co-staining for Olig2 and CC1 is required to assess the impact of platelet depletion on OPC differentiation.
- 5d) What about the effects on oligodendrocytes and/or remyelination at 14 dpi?
- 6) It is not clear why RNAseq analysis have been performed only on NSPCs co-culture with platelets since platelets affects also oligodendrocytes differentiation both in vitro and in vivo.

7) Regarding statistic:

The statistical analysis for the graph in Figure 4D appears to be incorrect: since there are two independent variables (condition: control vs. ipsilateral and genotype: WT vs. KO), a two-way ANOVA should be performed to account for both main effects and their potential interaction.

In some cases by the scatter plot graph it seems that only two values/biological samples have been meaned which is not acceptable (i.e. Fig1 I and H condition 1:40 and Fig1 H condition 1:1200; Fig 5D-G wt 5 and 14 dpi)

Minor point:

- 1) In Figure 4C, the dotted lines do not correspond to the outlined area.
- 2) Please express all the histological counting analysis in the graphs as n° of positive cells/mm²
- 3) Labels on graphs 3K and 3L are hardly visible and single values/biological samples are not shown

Version 1:

Reviewer comments:

Reviewer #1

(Remarks to the Author)

This paper is now suitable for publication in communications biology, as the authors have adequately addressed the previously raised issues.

Reviewer #2

(Remarks to the Author)

I thank the authors for addressing my comments. I also agree with them that proposing a mechanistic explanation of how platelets act on NSPCs/ OPCs "would be very speculative to formulate."

Reviewer #3

(Remarks to the Author)

Although I appreciate the authors' efforts in revising the manuscript, there are still several minor points that should be clarified and addressed. Please note that the line numbers refer to the marked-up PDF file:

- Line 115: The text reads "with approximately 10% of cells on day 3 co-expressing Ki67 (Fig. 1K, dark blue bar)." Should this instead refer to Fig. 1L?
- Figure 1Q: The axes should be labelled as "% of CC1+ or CC1/MBP double positive cells"
- Cell:platelet ratio calculation: I may be missing something here. If the affected area is 0.5 mm³, this would correspond to ~3 × 10⁶ cells, which is ~100 times more than the injected platelets (30,000). Please clarify this calculation.
- Figure 3E: the reported p-value is identical for both comparisons (p = 0.027 for "no PLTs vs. 1:75" and "no PLTs vs. 1:75 PKH26"). Could the authors confirm this? More generally, the presentation of statistics should be made consistent, since sometimes exact values are reported, while in other cases asterisks or hash marks are used.
- Figure 4M (PLP density graph): In both the text and the figure legend, this graph is incorrectly cited as Fig. 4J. In addition, the statistical analysis is missing from this graph and should be included.
- Figures related to SEZ (lines 320–330): Several figure citations in this section appear to be incorrect. Please check and correct.
- In the discussion the reference to Fig8 should be correct to Fig.7.
- Line 509: For clarity correct with "[...] to the increased presence of parenchymal platelets [...]"

Overall, with these minor corrections and clarifications, the manuscript would be suitable for publication.

Platelets regulate neural and oligodendroglial progenitors when infiltrating the brain parenchyma

Reviewer #1:

In this paper, the authors investigated the regulatory role of platelets for neural stem/progenitor cells (NSPCs) and oligodendrocyte progenitor cells (OPCs) both in vitro and in vivo, and they showed that platelets created a pro-stemness environment for NSPCs and enhanced the differentiation of OPCs.

This paper deserves publication in communications biology, if the authors properly address the following points.

Point 1. In figure 1E, the authors showed the effect of NSC-PLT ratio on the changes in % of SOX2+ cells normalized per “no Platelets”. They should show the percentage of SOX2+ cells in their NSPC preparation in the absence of platelets.

Response 1. *As we describe in the “results” text (lines 95-97 of the version **without tracked changes**) “Each biological NSPC sample was split into sub-cultures, exposed to different numbers of platelets (produced by the same platelet biological sample), at a ratio of “NSPC:platelets” that ranged between 1:40 and 1:1200, with a “no platelets” culture serving as the internal control.”*

This setup enabled us to repeat experiments at different labs and times (hence being sensitive to technical/ handling variations) and to provide data (shown in Figures 1E, F, G and Suppl. Figure 2) in a form that reveals more accurately and with less biological noise, the effects of platelets per biological sample. We then compared data using 1-way ANOVA, rather than some type of repeated measures analysis. This is explained in the Figure1 legend: “The value of each NSPC biological sample (formed by 1 to 3 technical replicates) is depicted with a circle and is normalized per the “no-platelets” culture of the same sample, which is at “1.0” and is indicated by the red, dotted line.”

Point 2. To my eyes, the shape of the blood vessel in the inset (C) of Figure 2H' is different from that in Figure 2H-C.

Response 2. *This was a very well-spotted error!! The blood vessel fragment was shown in inverted position in the magnification of the first submission. The Figure has been significantly revised and the respective fragment is now in Figure 2F3.*

Point 3. I would like to know what kind of cells were SOX2-/PCNA+ in the striatum injected with platelets.

Response 3. *We investigated the area of platelet injection further by co-immunostaining for PCNA and for oligodendrocyte lineage cells (immunopositive for Olig2), astrocytes (GFAP+) and microglial cells (Iba1+). Our results revealed that only a minimal fraction of all these cell types contributed to the pool of proliferating cells, with no difference observed between the site of platelet injection and of DMEM injection. Therefore, based on the evidence that the integrity of local blood vessels was compromised at the site of injection (e.g. with platelets finding their way*

Platelets regulate neural and oligodendroglial progenitors when infiltrating the brain parenchyma

in the vasculature, as shown in Sup Fig 3A) we expect that the majority of PCNA+ cells are blood-derived macrophages.

The data on the presence of oligodendroglial, astroglial and microglial cells at the areas of injections are presented in the revised Figure 3M (graph) and in the revised Sup Fig 3I-J (images). The “results” section has been amended accordingly and now reads as follows:

(starting on line 222) “To investigate more the identity of proliferating cells at the sites of injection in the striatum, we also immunostained for Olig2, GFAP, Iba1 and PCNA or phospho-Histone 3 (PH3) (Fig 3M, Suppl. Fig3I- J). We found only small numbers of mitotic Olig2+ and GFAP+ cells and no double positive Iba1/PH3 cells (Fig 3M). These data suggest that the majority of PCNA+ cells in both DMEM or platelet-injected striata were of peripheral origin (e.g. macrophages), a finding consistent with the presence of a compromised vascular integrity, as judged by the presence of injected platelets within local blood vessels (Suppl. Fig 3B, F, H).”

Point 4. Why didn't they detect any difference in the fraction of blood vessels with adherent CD41+ platelets between WT and Nbeal2-KO mice in non-injured hemisphere?

Response 4. *It is not clear why a fraction of platelets is found to adhere on the SEZ vasculature in homeostatic conditions. An explanation is possibly hiding in the specific characteristics of the SEZ blood vessels, as we describe them in the introduction (line 53): “The vasculature of the SEZ has a specialized architecture, with a blood vessel network that is denser and reaches the ventricular wall at shorter distances when compared to other periventricular areas^{12,13}, with low blood-flow rate^{12,14}, increased levels of leakiness¹⁵ and with endothelial cells that play a direct regulatory role on NSPCs¹⁶.”*

This is also highlighted in the revised text of the “results” section (line 268): “In the non-injured hemisphere, we detected a fraction of blood vessels with adherent CD41+ platelets, a phenomenon possibly facilitated by the low rate of blood flow¹⁴ observed in the SEZ, that was not different between the genotypes. However, in response to injury the fraction of blood vessel fragments with adherent platelets was significantly increased in wild type (WT) mice, a phenomenon that was absent from Nbeal2KO mice (Fig 4C-D).”

Most importantly, though, our analysis confirmed that after the demyelination challenge in the corpus callosum, the depleted pool of circulating platelets in Nbeal2 KO mice does not “support” the increased levels of aggregation in the SEZ vasculature, as observed in WT mice.

Point 5. They examined the effect of lysolecithin on WT mice, Nbeal2-KO mice, and chemically-induced thrombocytopenic mice, but they showed only the results in WT and Nbeal2-KO mice. I would like to know the results in chemically-induced thrombocytopenic mice.

Response 5. *We assume that the reviewer refers to the initial assessment of the levels of demyelination shown in Figure 4, as in the subsequent results section we include data from all experimental animals. We have now added the respective information for the analysis of myelin (PLP immunostaining) in mice with chemically depleted platelets, at 7dpl in the revised Figure 4I-M. Indeed, no differences were found in the levels of demyelination induced by lysolecithin between all experimental groups.*

Platelets regulate neural and oligodendroglial progenitors when infiltrating the brain parenchyma

The “results” text has been amended accordingly, and now reads as follows (line 257): “Lysolecithin, a demyelinating agent, was injected into the CC near to the SEZ. Tissue was analysed at 5-, 7- and 14-days post-lesion (dpl) in WT and in Nbeal2-KO mice and at 7dpl after chemically-induced thrombocytopenia (Figures 4A, 5A). The effects of lysolecithin were initially assessed by proteolipid protein (PLP), a key myelin component, immunoreactivity (Fig 4E-J), which was found to be reduced to 60% and 40% of its normal levels (the respective CC area in the non-lesioned hemisphere served as internal control) at 7dpl and 14dpl, respectively, in all experimental groups (Fig 4J). New myelin starts to be generated by OPC-derived oligodendrocytes after 14dpl and the area is fully remyelinated after 30 days42.”

Point 6. “Do” in Figure 4A should be “D0”.

Response 6. *Another acute observation, which we have corrected in revised Figures 4A & 5A.*

Point 7. In line 307, they wrote “SOX2 is also expressed by ependymal cells and reactive astrocytes”.

Response 7. *We are not sure what the concern is here. Indeed, SOX2 is expressed by ependymal cells and reactive astrocytes, but this -possibly unnecessary- clarification has been removed.*

Point 8. In line 310, isn't (Fig5F-G) proper, instead of (Fig5D-G)?

Response 8. *The reference to Fig5D-G was covering all the results (total cells, PCNA+ cells, Olig2+ cells and Dcx+ cells) that were provided just prior. To make the text clearer we inserted additional references to the Figure's panels.*

The revised text reads as follows (starting on line 308): “We quantified the presence of PCNA, DCX and OLIG2 immunopositive cells in the SEZ (Fig 5B-C). All PCNA+ cells within the niche co-express the transcription factor SOX213 (Fig 5D); thus, they were considered to be NSPCs. Focal demyelination in the CC did not affect the total cell density in the WT SEZ (Fig. 5E, black bars), although it led to a transient, significant, increase in the percentage of PCNA+ cells at 5dpl (Fig. 5E); thus, indicating that the injury affected the niche. The pools of OLIG2+ and DCX+ progenitors were not affected (Fig 5D-G, black bars). Long-term thrombocytopenia had no effects in the SEZ (Fig. 5E- H, cyan bars). However, short-term thrombocytopenia led to a significant increase in the percentage of OLIG2+ cells, both in the control SEZ and at 7dpl, compared to the WT and the Nbeal2KO mice, alike (Fig 5H, magenta bar). The pool of DCX+ cells remained unaffected by thrombocytopenia (Fig 5G). Next, we focused our analysis on the site of lesion in the CC (Fig. 5I-N). In WT mice, demyelination resulted in a significant increase in the total cell density at 7dpl and in the percentage of PCNA+ cells- at 5dpl and 7dpl (Fig 5I- J), as expected due to the recruitment of mitotically active macrophages and microglial cells. It also led to the significant decrease in the percentage of OLIG2+ cells at 5dpl and 7dpl (Fig 5K), the cell lineage targeted by lysolecithin, as also revealed by PLP staining (Fig. 4E- J).”

Point 9. In line 327, “form” should be “from”.

Response 9. *The text has been corrected (revised text, line 316).*

Platelets regulate neural and oligodendroglial progenitors when infiltrating the brain parenchyma

Point 10. Their RNA-seq analysis revealed the activity of PDGF on NSPCs cocultured with platelets. PDGF is generally considered to be released from activated platelets. They should show the percentage of activated (CD62P+) platelets in their NSPC-platelet co-cultures.

Response 10. *The information requested by the reviewer is already included in the submitted manuscript (Suppl. Figure 1G), along with two other bits of evidence that offer an insight on the activation status of platelets in the co-culture experiments:*

First, that during isolation and washing, platelets were handled in such a way that they were kept inactivated. This was assessed by FACS (Suppl. Figure 1A- D) or by immunocytochemistry (Suppl. Figure 1F) for CD41 and CD62P, on platelet preparations.

Second, that after three days in co-culture with NSPCs, judging by immunocytochemistry, we could detect activated platelets (Suppl. Figure 1H).

Third, as the reviewer highlighted, that the RNAseq analysis revealed that the presence of platelets resulted in the transcriptional activation of PDGF-related pathways (Figure 6F).

It is technically very difficult to assess (accurately) with the resolution provided by immunocytochemistry, the levels of activation of such small cells within the cell cultures. To increase the depth of the analysis, in the revised manuscript we have included Suppl. Figure 1I. This is a graph reporting mitochondrial activity, measured with the MTT assay, in pure platelet cultures, in differentiation or proliferation media for up to 5 days. Mitochondrial activity is considered to be an indication of the viability of platelets and its loss the result of the post-activation exhaustion of platelets (<https://doi.org/10.1038/s41598-025-91181-y>).

This analysis revealed that: a) the activation of platelets is very fast (it happens almost in full even on the first day in vitro) for NSPC:platelets ratios up to 1:40; b) that by the third day in culture mitochondrial activity has dropped by more than 50%; c) that by the fifth day in culture, mitochondrial activity is almost lost.

The additional information is now included in the results section, starting on line 101: “By assessing the percentage of CD41+ platelets co-expressing CD62P, it was revealed that platelets became activated in vitro and that in the presence of NSPCs their activation was significantly lower than when cultured on their own (Suppl Fig 1H), indicating that these cell groups were interacting with each other. In a separate approach, measuring mitochondrial activity with the MTT assay⁴³, we were able to confirm that platelets became activated when kept in NSPC media, with their activation being very fast in low densities (relevant to NSPC:platelet ratios up to 1:40) and with platelets becoming fully exhausted after 5 days in culture (Suppl Fig 1I).”

Point 11. Conditioned media obtained at the end of 3-day NSPC-platelet (1:75) co-cultures did not lead to an increase in the number of SOX2+ cells. They should explain this.

Response 11. *The specific experiment revealed that the induction of SOX2 expression was dependent on direct cell contact (and that conditioned medium was toxic to NSPCs). These differential effects exerted by direct platelet:NSPC contact and by platelet-conditioned medium are an interesting result that we did not investigate much further. Nevertheless, we now discuss its implications as part of the question of whether platelets can affect NSPCs only in direct proximity.*

Platelets regulate neural and oligodendroglial progenitors when infiltrating the brain parenchyma

The revised text now reads as follows (line 502, with references 64 and 65 added to the manuscript): "Our data, showing that platelet-conditioned medium exerts different effects as compared to the direct platelet:NSPC contact (Suppl. Fig 3G, H) also highlight that the activity of platelets is not mediated only by diffusible factors, but is also dependent on cell:cell, mechanotransduction, mechanisms, because platelets can sense and respond to changes in their microenvironment⁶⁴, possibly via integrins^{30,58,65}."

Platelets regulate neural and oligodendroglial progenitors when infiltrating the brain parenchyma**Reviewer #2 (Remarks to the Author):**

The manuscript titled "Platelets regulate neural and oligodendroglial progenitors when infiltrating the brain parenchyma" by Dimitriou et al. explores the role of platelets in modulating the behavior of neural stem/progenitor cells (NSPCs) and oligodendrocyte progenitor cells (OPCs) within the brain. The conclusions from this study emphasize the supportive and complex roles platelets can play in neuro-regenerative strategies, particularly in conditions where platelets infiltrate the brain parenchyma due to vascular damage or pathological conditions. The findings are interesting and suggest significant translational aspects. However, certain areas need clarification and improvement to enhance the scientific rigor, presentation, and broader relevance. Special attention should be given to the quality and readability of the figures as outlined below.

General comments:

Point 1. Unless justified, I recommend to use consistent terminology throughout the manuscript. For instance, 'NSPC' and 'NSC' are used interchangeably. Stick to one term (e.g., "NSPC") and ensure it is applied consistently (e.g., Fig 1, Fig S3).

Response 1. *We have gone through the text and this inconsistency has been corrected, mainly in Figures reporting co-cultures.*

Point 2. I suggest to directly label specific figures or panels within figures to explicitly indicate the experimental model that has been used to generate the data. This will greatly help readers to follow the experimental design and the corresponding results.

Response 2. *We have gone through the figures and we have added schematic illustrations that visualise and explain the experiments reported (e.g. Fig. 1, 2, 3, 4, 5 & Suppl. Fig 3).*

Point 3. A more detailed explanation of how platelets influence NSPC and OPC behaviors could be proposed. For instance incorporating a pathway or model diagram would enhance understanding and the scientific impact of the manuscript.

Response 2. *We have worked on the available data to assess if we could propose a clear and well-supported mechanistic explanation of how platelets act on NSPCs/ OPCs. However, apart from the description of the many ways, and of the conditions in which, platelets can affect the behavior of NSPCs and of OPCs, our data on the molecular level are insufficient at this stage. To be as analytical as possible we included a detailed summary of the genes of interest that emerged from the RNAseq experiment.*

In our opinion, it would be very speculative to formulate and suggest anything more and we chose to avoid such an attempt.

Point 4. The controls used, especially during in vivo experiments, should be more clearly presented to reinforce the strength of the conclusions.

Platelets regulate neural and oligodendroglial progenitors when infiltrating the brain parenchyma

Response 4. *We have revised the “results” text in the section where we report the descriptive analysis of complementary injury models (starting on line 166), to clarify the three different experiments and the key observations.*

We have also made focused changes in the text, and we have updated the Figures with the relevant schematic illustrations. We hope the different experimental strategies are now more accessible to readers.

Point 5. The manuscript has a few typos that should be corrected: “neuramnidase”, perturbed -> perturbed (discussion); “in re” (Fig 1 legend); “exeperimental” (fig 5 legend); “form” --- > “from” (Methods); CD1+ ---- > CD41+ (Fig 2H).

Response 5. *We have gone through the text and have corrected the mistakes.*

Point 6. « This work advances the potential to use this naturally unlimited, and suitable for autologous use, source of cells in regenerative biomedicine » in the introduction and b) “their unlimited availability for autologous and heterologous use” in the discussion are statements that deserve further exploration and discussion with some brief explanations on how this can be achieved with a reference to current developments seen in the use of allogenic platelets materials from healthy donors for brain administration and treatment of CNS disorders.

Response 6. *In the “conclusion” the text was amended to reflect -in a clearer way- the prospect of transiently manipulating the numbers of circulating platelets, to achieve a regulation of their presence in areas of degeneration, if that proves beneficial. Moreover, we also changed the text to highlight that autologous preparations of platelets can be used to facilitate cell-transplantation strategies, since platelets can be safely collected in high numbers and in high frequency.*

The text reads as follows (starting on line 577): “This activity of platelets could be a novel target as part of neuroprotective and neuroregenerative strategies, especially because numbers of circulating platelets can be manipulated pharmacologically, as well as with plateletpheresis (removal of platelets) or platelet transfusions from donors. Moreover, platelets can be a valuable cell source for the improvement of NSPC/OPC-based transplantation strategies, either by priming cells before grafting or in co-transplantations. This option is facilitated by the easiness of collecting platelets for autologous use since platelet donations can be safely performed as often as every 2 weeks.”

Specific comments:

Point 7 on M&M: (a) Platelet isolation: what kind of anticoagulant was used? What were the volumes processed? Platelets are not “redissolved” but “resuspended”.

Response 7 on M&M: *The details of the platelet isolation protocol were provided in the Supplemental Material, but additional key information has now been provided in the main text “M&M”, with the specific section reading as follows:*

(line 606): “Platelets were isolated from the vena cava inferior⁸⁹, by collecting approximately 1ml of whole blood in tubes containing EDTA buffer (0.5M, pH 8) as anticoagulant and subsequently separating and washing the platelet rich-plasma fraction (PRP) using Tyrode’s buffer. Apyrase and PGE1 were added to the PRP in order to avoid platelet activation and PRP was then centrifugated. At the end, platelets were resuspended in NSPC or OPC culture medium.”

Platelets regulate neural and oligodendroglial progenitors when infiltrating the brain parenchyma

The relevant section in the Supplemental Material reads as follows:

“Platelets were isolated from the inferior vena cava¹. Briefly, mice were given an i.p. overdose of ketamine (200µg per g of body weight). The thorax was exposed and approximately 1mL of blood was collected using a 27G needle and transferred into tubes with 100µl of EDTA (0.5M, pH 8) solution. The blood was transferred into 5mL RIA/FACS polystyrene tubes with the addition of 1,5 x of total volume Tyrodes buffer (134mM NaCl, 2.9mM KCl, 0.34mM Na₂HPO₄·2H₂O, 12mM NaHCO₃, 20mM HEPES, 1mM MgCl₂·6H₂O and 5mM D-Glucose anhydrous). Tubes were centrifuged (100g, 20min) and the platelet rich-plasma layer was collected, in the presence of 0.5u/ml of apyrase (A6410, Merck Sigma- Aldrich) and 50ng/ml of PGE1 (P5515, Merck Sigma- Aldrich). After another centrifugation (1000g, 10min) the pellet was resuspended in NSPC or OPC culture medium.”

Point 8 on Figure legends: Please ensure that all statistical methods are clearly described in the legends and present results with consistent and appropriate indicators of statistical significance (e.g., p-values, error bars). Also please specify how many independent experiments were conducted. N (independent experiments) = ?

Response to point 8 on Figure legends:

We have revisited all figures and graphs and we have made every effort to be as transparent as possible by showing all individual experimental values in the scatter plots, or by referencing the number of biological samples. In the revised manuscript we are more detailed on presenting the statistical tools used in each experiment and to describe the use of the relevant symbols (stars, lines etc).

Point 9 on Fig 1 and others: It is difficult if not impossible to see all data points and in some instances the number of data points seems insufficient and not able to support the calculated averages. In addition it is stated that “The value of each experiment (biological sample) is depicted with a circle and is normalized per the “no-platelets” culture of the same sample, which is at “1.0” and is indicated by the red, dotted line”: why the authors does not show the individual data points for the “no-platelets” condition in addition to the dotted line for transparency of the data?

Response 9. *Part of this point was addressed just above, and another part is relevant to reviewer's #1 point 1, that we have addressed in a similar way.*

As described in the “results” text (lines 95-97) “Each biological NSPC sample was split into sub-cultures, exposed to different numbers of platelets (produced by the same platelet biological sample), at a ratio of “NSPC:platelets” that ranged between 1:40 and 1:1200, with a “no platelets” culture serving as the internal control.”

This setup enabled us to repeat experiments at different labs and times (hence being sensitive to technical/ handling variations) and to provide data (shown in Figures 1E, F, G and Suppl. Figure 2) in a form that reveals more accurately and with less biological noise, the effects of platelets per biological sample. We then compared data using 1-way ANOVA, rather than some type of repeated measures analysis.

Platelets regulate neural and oligodendroglial progenitors when infiltrating the brain parenchyma

This is explained in the revised Figure 1 legend: "The value of each NSPC biological sample (formed by 1 to 3 technical replicates) is depicted with a circle and is normalized per the "no-platelets" culture of the same sample, which is at "1.0" and is indicated by the red, dotted line."

If the reviewer prefers to have the "no-platelets" conditions shown in the form of a separate bar (with a set value of 1 and without error bars, as done in Suppl. Fig 3G, H where there are only three experimental conditions to be compared) instead of the horizontal red, dotted, line (that we find to be visually more informative) we are more than happy to accommodate this.

We have revisited all figures and graphs and we have made every effort to be as transparent as possible by showing all individual experimental values in the scatter plots, or by referencing the number of biological samples. A restricted number of time-points per experimental group (in vivo and in vitro) include sub-optimal "n" numbers, but this does not affect the key results. It must be noted that in each experiment we first produced the overall p-value using ANOV Analyses and only if this was indicating the presence of statistical differences did we perform the Tukey post-hoc analyses.

Point 10 on Fig 1D-E: Please clarify whether the increased Sox2 expression without DAPI changes indicates increased Sox2 protein levels without changes in cell number or nuclear size. Specify whether this signifies "stemness" and describe how Sox2+ cells were quantified (e.g., area- or cell count-based). One condition has a large error bar: is there any experimental explanation.

Response to Point 10: *This is experimental work that involves many approaches, each one of which could be analysed in further depth. SOX2+ expression was assessed as number of immunopositive nuclei per number of total nuclei, both in vitro and in vivo, without assessing the levels of immunoreactivity or the size of the nuclei; thus, we did not assess the quantity of the protein. The simplest explanation for the increase in the number of SOX2+ cells, especially in the absence of a concurrent change in the total number of cells (as judged by the number of DAPI+ nuclei) in the co-cultures is that in a fraction of cells the expression of SOX2 is either not switched-off or is re-induced. In the in vivo, platelet-injection experiment, the emergence of SOX2+ cells is interpreted as an indication of the emergence of cells with NSC character, as we consider each SOX2+ cell to be a cell with neural stem cell properties.*

These conclusions are noted in the text:

(line 139) "Together, our in vitro data revealed that platelets are able to create a pro-stemness microenvironment for NSPCs, enhancing (maintaining or inducing) the expression of SOX2 and supporting mitosis, and to promote differentiation of parenchymal OPCs."

(line 230) "The observation that the direct injection of platelets in the striatum results in the emergence of SOX2+ cells, a fraction of which are mitotic SOX2+ cells and are not co-expressing Olig2 or GFAP leads to the conclusion that platelets induce the recruitment of parenchyma neural progenitors."

Point 11 on Fig 1F: Clarify the discrepancy between the 3.5-fold Ki67 increase and the unchanged DAPI staining. This requires further clarity.

Response 11: *As shown in Fig 1H, the percentage of Ki67+ cells in the high-platelet co-cultures is around 10%. The presence of such a low fraction of mitotic NSPCs, for an unknown time-frame (we do not have data on what happens during days 1 and 2 in the co-cultures, but as shown in Fig*

Platelets regulate neural and oligodendroglial progenitors when infiltrating the brain parenchyma

1H, under proliferation conditions the percentage of Ki67+ NSPCs falls from approximately 20% on day 1 to 15% on day 3) is compatible with the absence of a significant change in the total number of cells in the culture. This is also with the levels of cell death being un-investigated.

The text was updated on line 112, to reflect this point: "Levels of proliferation, that are markedly low in these differentiation conditions, showed a dichotomous effect: the 1:40 NSPCs:platelets ratio resulted in significantly reduced levels of Ki67, while the 1:1200 ratio led to a 3.5 times, significant, increase (Fig 1G), reaching approximately the 10% of cells on day 3 (Fig 1H, dark blue bar)."

Point 12 on Fig 3F-L: the figure organization is confusing and it is recommended to reorganize the panels for improved readability by grouping images with similar experimental conditions (e.g., group striatum/SEZ sites and marker series).

...and specifically for Fig 3H, include a control image with the same frame selection as the injection site and provide high-magnification images of both conditions to enhance clarity.

Response 12: Figure 3 has been re-organised to show the experimental results in a clearer and more coherent way. We have updated the images by adding high magnification data from both hemispheres.

Point 13 on Fig 4D: please define "b.v." to avoid confusion.

Response 13: The title of the graph in Fig 4D has been revised.

Point 14 on Fig 5A-C. To enhance the clarity of the results, a representative image of the dysmyelinated hemisphere in WT mice for direct comparison with control hemisphere should be included.

...and more specifically on Fig 5F Please add statistical significance indicators.

Response 14: For purposes of efficiency in the use of space, as this is an extensive manuscript, we strived to provide only the necessary visual data. The effects of demyelination are shown in detail (especially after the inclusion of higher magnification images in the revised Fig 4E- I) in Fig 4. Fig 5 was made less dense, with immunostaining images being moved to Suppl. Figure 2. In these images we only show the pattern of the immunostainings we worked with, relying on the graphs to report the results.

Point 15 on Fig S1E, F: Can the resolution of the images be improved ? Clarify the presence of erythrocytes in Fig S1E if applicable.

Response 15: To visualise and to count platelets manually, using a haematocytometer, we had to focus on the layer of the much smaller than the erythrocytes platelets. This is why red blood cells are shown out of focus, although being clearly of a larger size. This detail has been now added to the legend of Suppl Fig 1. Unfortunately, these images that have been obtained by a tissue-culture grade inverted microscope are of lower resolution and cannot be improved much further.

We note that additional images of co-cultures have been included in Figure 1 and Suppl. Figure 1.

Platelets regulate neural and oligodendroglial progenitors when infiltrating the brain parenchyma

Platelets regulate neural and oligodendroglial progenitors when infiltrating the brain parenchyma**Reviewer #3 (Remarks to the Author):**

In this paper by Dimitriou et al., the authors studied the effects of platelets on neural and oligodendroglial progenitors both in vitro and in vivo. The authors suggested that when co-cultured with NSPCs or OPCs, platelets were able to create a pro-stemness microenvironment and promote OPC differentiation. In vivo, thrombocytopenia without platelet tissue infiltration in Nbeal2 knockout mice did not affect the brain's progenitors. However, when platelets were injected directly into the striatal parenchyma or extravasated in severely thrombocytopenic mice, there was an increase in Olig2+ and Sox2+ cells.

The manuscript addresses a potentially interesting topic, however, the presentation of the data is not completely clear, and the narrative structure could be significantly improved. I recommend reviewing the text to create a more logical flow and providing additional clarification or context for the presented data.

Major points:

Point 1. It is not clear how the proportion of NSPCs/OPCs to platelets was calculated and chosen. Does the ratio of "NSPC:platelets" 1:1000 mean that platelets are 1000 times more abundant than the other cells? In a previous paper published by the same group (Philp et al., eLife 2023;12:RP91757. DOI: (<https://doi.org/10.7554/eLife.91757>)(<https://doi.org/10.7554/eLife.91757>), OPCs were subjected to significantly lower concentrations of platelets (1%, 5%, and 10%). Please clarify.

Response 1. *As explained in the revised "Materials and Methods" section, "Co-cultures were performed in multi-well plates, with cells grown on PDL-coated glass coverslips. NSPCs or OPCs were plated the day before the addition of platelets. A range of NSPC or OPC:platelet ratios were used (1:40, 1:75, 1:400, 1:1200, 1:3000) with the numbers of NSPCs/OPCs and platelets calculated with the use of a haematocytometer".*

The reviewer is right to consider that this was the actual ratio between the number of NSPCs or OPCs and Platelets.

In Philp et al., the methodology is somewhat different, reflecting the historical differences in the way platelets were calculated and used between the Rivera lab (that led the work reported in the Elife paper) and the Kazanis lab (that led the work reported here).

The way the Rivera lab operates, washed platelets are suspended and stored in a density of 10^6 platelets/ μ l. Thus, in a 24-well plate seeded with 7000 OPCs that are cultured in 250 μ l of media, the ratio OPCs:platelets was 1:350 (1%), 1:1750 (5%) and 1:3500 (10%), which is not that different from the ratios we tested.

Point 2. In Figure 2, a quantification of the observations would enhance the significance of the data.

Response 2. *The data presented in Figure 2 and described in the respective, second, section of "Results" are analysed only in a dichotomous way: do we observe platelets only within the*

Platelets regulate neural and oligodendroglial progenitors when infiltrating the brain parenchyma

vasculature, or also having infiltrated the parenchyma? This was a key question of the experimental project

Any attempt to quantify the presence of platelets, within and (even more) outside the blood vessels, is extremely challenging at the technical level and would be, in terms of the biological question, outside the scope of this experimental work.

Point 3. I have some concerns regarding how the data on the effects of platelets on OPC differentiation have been presented. Firstly, the authors stated: “the numbers of CC1+ and MBP+ cells were also found to be significantly increased in the presence of 3000 platelets per OPC (Fig. 1o)” [line 127]. However, the graph in Fig. 1o only shows CC1+ and CC1-MBP+ cells, and it is unclear whether these are expressed as percentages or absolute numbers (as indicated in the figure legend, line 165: “Graphs showing [...] the number of CC1+ and MBP+ cells per optical field”). To improve clarity and consistency, please modify this graph to present the data as:

- The percentage of CC1+ and MBP+ cells over the total number of cells.
- The percentage of CC1+ - MBP+ double-positive cells over the total number of CC1+ cells.

Response 3. *The wording in the legend of Figure 1 was wrong. As shown on the respective graph (Fig 1O) what we present is the percentage of CC1+ and of CC1+/MBP+ cells. The legend has been corrected. It should be also clarified that all MBP+ cells were co-expressing CC1; hence, we do not present data for an MBP+ but CC1- cell population.*

Point 4. How was the number of stereotactically injected platelets determined? Is the ratio between cells and platelets similar to that used in the in vitro experiments? Additionally, the effects on the oligodendrocyte population should be investigated.

Response 4. *Yes, we tried to align together the different experimental approaches, with a cell:platelets ratio around 1:75 being used to test conditioned media, to perform the RNAseq analysis and the effects of intracerebral platelet injections.*

To clarify this point, we changed the relevant text in the “results” section. The text on line 209 now reads as follows: “Subsequently, approximately 30000 platelets were unilaterally injected into the striatum, a structure harboring latent neural progenitors in the mouse brain²⁵ (Suppl. Fig 3A-J) or at the SEZ dorsal horn (Suppl. Fig 3K-L), with DMEM injected into the same areas of the contralateral hemisphere. Based on tissue cell-densities calculated in previous experimental work⁴² and on further pilot calculations, we estimated an average cell density of 6×10^6 cells/mm³ (total cells, including neurons, glia and endothelial cells). We also estimated that the injection of 1 μ l of platelets/ DMEM would directly affect a volume of 0.5mm³. Thus, the injection of 30000 platelets would result in a cell:platelet ratio of 1:100.”

In the revised manuscript we have also included an analysis of other proliferating cell populations at the areas of platelet/ DMEM injections, to assess the profile of mitotic cells (this is relevant to the 3rd point of reviewer#1).

We performed co-immunostaining for PCNA and for oligodendrocyte lineage cells (immunopositive for Olig2), astrocytes (GFAP+) and microglial cells (Iba1+). Our results revealed that only a minimal fraction of all these cell types contributed to the pool of proliferating cells, with no difference observed between the site of platelet injection and of DMEM injection.

Platelets regulate neural and oligodendroglial progenitors when infiltrating the brain parenchyma

The data on the presence of oligodendroglial, astroglial and microglial cells at the areas of injections are presented in the revised Figure 3M (graph) and in the revised Sup Fig 3I-J (images). The "results" section has been amended accordingly and now reads as follows:

(starting on line 222) "To investigate more the identity of proliferating cells at the sites of injection in the striatum, we also immunostained for Olig2, GFAP, Iba1 and PCNA or phospho-Histone 3 (PH3) (Fig 3M, Suppl. Fig3I- J). We found only small numbers of mitotic Olig2+ and GFAP+ cells and no double positive Iba1/PH3 cells (Fig 3M). These data suggest that the majority of PCNA+ cells in both DMEM or platelet-injected striata were of peripheral origin (e.g. macrophages), a finding consistent with the presence of a compromised vascular integrity, as judged by the presence of injected platelets within local blood vessels (Suppl. Fig 3B, F, H)."

Point 5. The in vivo results regarding the effects of long-term and short-term thrombocytopenia are unclear.

- 5a) Please include representative images in Figure 5 for at least the condition where statistical significance is observed, highlighting the area where cells were counted.
- 5b) Results should be expressed as the number of positive cells per mm² (n° positive cells/mm²).
- 5c) The effects on oligodendrocytes are poorly characterized. At the very least, a co-staining for Olig2 and CC1 is required to assess the impact of platelet depletion on OPC differentiation.
- 5d) What about the effects on oligodendrocytes and/or remyelination at 14 dpi?

Response 5.

5a) *This is a manuscript in which we report many complementary experimental approaches. We have tried to incorporate as much visual evidence as possible, aiming at demonstrating how immunostainings looked like and what the key areas we analysed were within the confined, available space. The results are accurately provided in the graphs.*

The areas of analysis (demyelination) are shown in Figure 4E- H, with higher magnifications after immunostaining for PLP and CNPase in Figure 4I-L and in Suppl. Figure 4D-G (this a new addition), respectively. Additional representative images of how the different markers looked like in our immunohistochemical analyses, with annotations of the areas analysed (corpus callosum and the dorsal SEZ), are provided in Suppl. Figure 4A-C.

5b) *When histological and immunohistological comparisons are performed across different animal models (wild-type mice with or without depletion of platelets, Nbeal2 knockout mice), at different laboratory setting and animal facilities, there is always an element of methodological variation in the way the tissue is perfused and post-fixed. This incorporates differences that can affect significantly measurements of cell density (which depends on measuring the volume of the tissue). If in some animals the tissue has shrunk differently during the process, this could introduce significant differences. Therefore, if the experiments have not been performed in a strictly controlled way, with the aim of producing very comparable tissue volumes, cell density measurements are not very reliable (refer also to the work of Suzana Herculano-Houzel on the cell density in the human brain (<https://doi.org/10.1002/cne.24040>). On the other hand, the calculation of cell percentages is less dependent on volume variability and this is why we chose to show this information. Cell percentages provide a measure of the representation of the*

Platelets regulate neural and oligodendroglial progenitors when infiltrating the brain parenchyma

different cell-types in the areas of investigation which is an important readout of the response of the tissue and of different cell-groups in experimental manipulation.

Nevertheless, if the reviewer is of the opinion that cell densities should be included, we are happy to provide them as additional supplemental material.

5c) The effects of demyelination in the different experimental groups (WT with or without depletion of platelets and Nbeal2-KO mice) are assessed in three ways. Firstly, by looking at the state of myelin, using immunostaining for PLP (Figure 4E- L). Secondly by looking at the pool of Olig2+ cells and even more specifically on OPCs, by co-immunostaining for Olig2 and PCNA (Figure 4I, J). Thirdly, we have now added an analysis of CNPase+ cells, to visualise maturing oligodendrocytes (Figure 4K and Suppl. Figure 4D- G) in a single-cell resolution that is not provided by MBP and PLP immunostaining.

Overall, our data revealed that the transient, severe, depletion of platelets was the only condition leading a significant increase in the total numbers of Olig2+ cells, but without significant effects on the mitotic fraction of OPCs, on the levels of PLP immunoreactivity (providing an estimate of the total quantity of myelin) and on the subpopulation of CNPase+ maturing oligodendrocytes. The oligodendroglial cell lineage includes many different stages of maturation, from that of the oligodendrocyte progenitor, up to a fully mature, myelinating, oligodendrocyte. Additional, and more focused, analysis is necessary to identify the stage in the lineage that is mostly affected by the depletion of platelets.

The “conclusions” text was modified to highlight this point (line 572): “Our data revealed that the cells of the oligodendroglial lineage are especially responsive to platelets. Their differentiation was enhanced by the presence of platelets in vitro, and the total number of Olig2+ cells remained significantly increased after demyelination. Our further analysis using different markers of the lineage, such as PLP and CNPase, and the co-staining for Olig2 and PCNA did not allow the identification of the specific maturation stage that is directly affected by platelets and this needs to be addressed in the future.”

5d) Regrettably, for reasons beyond our control, there was no option to analyse the corpus callosum at the 14dpl, similarly to the analysis performed for the other time-points and for the 14dpl time-point in the SEZ.

Point 6. It is not clear why RNAseq analysis have been performed only on NSPCs co-culture with platelets since platelets affects also oligodendrocytes differentiation both in vitro and in vivo.

Response 6. *This is another limitation of the study, due to restrictions in funding and resources.*

Point 7. Regarding statistic:

The statistical analysis for the graph in Figure 4D appears to be incorrect: since there are two independent variables (condition: control vs. ipsilateral and genotype: WT vs. KO), a two-way ANOVA should be performed to account for both main effects and their potential interaction. In same cases by the scatter plot graph it seems that only two values/biological samples have been meaned which is not acceptable (i.e. Fig1 I and H condition 1:40 and Fig1 H condition 1:1200; Fig 5D-G wt 5 and 14 dpl)

Platelets regulate neural and oligodendroglial progenitors when infiltrating the brain parenchyma

Response 7. *The reviewer is correct about the statistical analysis presented in Figure 4; although a 2-way ANOVA was performed, followed by the Tukey post-hoc analysis, by mistake it was grouped with the 1-way ANOVA performed for data shown in Figure 4B. The legend has been corrected.*

We have revisited all figures and graphs and we have made every effort to be as transparent as possible by showing all individual experimental values in the scatter plots, or by referencing the number of biological samples. A restricted number of time-points per experimental group (in vivo and in vitro) include sub-optimal “n” numbers, but this does not affect the key results. It must be noted that in each experiment we first produced the overall p-value using ANOV Analyses and only if this was indicating the presence of statistical differences did we perform the Tukey post-hoc analyses.

Minor point:

Point 8. In Figure 4C, the dotted lines do not correspond to the outlined area.

Response 8. *The figure has been revised and this error has been corrected.*

Point 9. Please express all the histological counting analysis in the graphs as n° of positive cells/mm²

Response 9. *Please see our response to your previous point 5b.*

Point 10. Labels on graphs 3K and 3L are hardly visible and single values/biological samples are not shown

Response 10. *Figure 3 has been revised.*

Platelets regulate neural and oligodendroglial progenitors when infiltrating the brain parenchyma**Reviewer #3 (Points raised about 1st revision of manuscript):**

Point 1: Line 115: The text reads “with approximately 10% of cells on day 3 co-expressing Ki67 (Fig. 1K, dark blue bar).” Should this instead refer to Fig. 1L?

Point 1 response:

The reference to the Figure has been corrected.

Point 2: Figure 1Q: The axes should be labelled as “% of CC1+ or CC1/MBP double positive cells”

Point 2 response:

The axis has been corrected.

Point 3: Cell:platelet ratio calculation: I may be missing something here. If the affected area is 0.5 mm³, this would correspond to $\sim 3 \times 10^6$ cells, which is ~ 100 times more than the injected platelets (30,000). Please clarify this calculation.

Point 3 response:

This is an important detail that was not addressed correctly, as we offered only part of the information. The key, missing, point was that the choice of injecting 30000 platelets was based on pilot studies revealing that this was the maximum number of platelets we could concentrate in 1 μ l of medium without, visually, observing activation (e.g. clotting). We, next, calculated the ratio of cells:platelets which, as the reviewer rightly mentioned was 100:1 and not 1:100, as was written in the manuscript.

The text has been amended to incorporate all this information:

(lines 212 -224) *“The injection dose was selected as it constituted the maximum number of platelets that could be concentrated within 1 μ l of medium without visual signs of activation (e.g. clotting). Based on tissue cell-densities calculated in previous experimental work⁴² and on further pilot calculations, we estimated an average cell density of 6×10^6 cells/mm³ (total cells, including neurons, glia and endothelial cells). Considering that the injection of 1 μ l of platelets/ DMEM could directly affect a maximum volume of 0.5mm³, the injection of 30000 platelets would result in a cell:platelet ratio of 100:1, although this ratio would be closer to the ratios tested in vitro (such as 1:100) more proximal to the site of injection.”*

Point 4: Figure 3E: the reported p-value is identical for both comparisons (p = 0.027 for “no PLTs vs. 1:75” and “no PLTs vs. 1:75 PKH26”). Could the authors confirm this? More generally, the presentation of statistics should be made consistent, since sometimes exact values are reported, while in other cases asterisks or hash marks are used.

Point 4 response:

The p-value indicated in Figure 3E was that of the ANOVA (on the effect of the culture conditions); hence, the single value. The post-hoc p-values have been added.

We also added the p-values in the graph of Figure 1K, where we only showing asterisks.

Platelets regulate neural and oligodendroglial progenitors when infiltrating the brain parenchyma

Point 5: Figure 4M (PLP density graph): In both the text and the figure legend, this graph is incorrectly cited as Fig. 4J. In addition, the statistical analysis is missing from this graph and should be included.

Point 5 response:

In-text and in-legend citations have been corrected.

We have now added the main statistics on the graph and have included more details in the legend. We do not show the detailed post-hoc analyses, for clarity and simplicity. We show: a) the overall ANOVA p-value for treatment and the lack of overall significance for time. We do not show the lack of overall significance for the experimental group, which is mentioned in the respective text (line 283).

Point 6: Figures related to SEZ (lines 320–330): Several figure citations in this section appear to be incorrect. Please check and correct.

Point 6 response:

Citations to figure 5B- E have been corrected.

Point 7: In the discussion the reference to Fig8 should be correct to Fig.7.

Point 7 response:

Citations to figure 7 have been corrected.

Point 8: Line 509: For clarity correct with“ [...] to the increased presence of parenchymal platelets [...]”.

Point 8 response:

The text has been amended.